



# Retrieving microphysical properties of concurrent pristine ice and snow using polarimetric radar observations

Nicholas J. Kedzuf[1], J. Christine Chiu[1], V. Chandrasekar[2], Sounak Biswas[2], Shashank S. Joshil[2], Yinghui Lu[3], Peter Jan van Leeuwen[1,4], Christopher Westbrook[4], Yann Blanchard[5], Sebastian O'Shea[6]

[1]Department of Atmospheric Science, Colorado State University, Fort Collins, CO, 80523, USA
[2]Department of Electrical and Computer Engineering, Colorado State University, Fort Collins, CO, 80523, USA
[3]Department of Meteorology and Atmospheric Science, and Center for Advanced Data Assimilation and Predictability Techniques, The Pennsylvania State University, University Park, PA 16802, USA
[4]Department of Meteorology, University of Reading, Reading, RG6 6BB, UK
[5]ESCER Centre, Department of Earth and Atmospheric Sciences, University of Quebec at Montreal, Montreal, Quebec, postcode, Canada
[6]Department of Earth and Environmental Sciences, University of Manchester, Manchester, M13 9PL, UK

*Correspondence to*: J. Christine Chiu (Christine.Chiu@colostate.edu)

**Abstract.** Ice and mixed phase clouds play a key role in our climate system, because of their strong controls on global precipitation and radiation budget. Their microphysical properties have been characterized commonly by polarimetric radar measurements. However, there remains a lack of robust estimates of microphysical properties of concurrent pristine ice and

aggregates, because larger snow aggregates often dominate the radar signal and mask contributions of smaller pristine ice crystals. This paper presents a new method that separates the scattering signals of pristine ice embedded in snow aggregates in scanning polarimetric radar observations and retrieves their respective abundances and sizes for the first time. This method, dubbed ENCORE-ice, is built on an iterative stochastic ensemble retrieval framework. It provides number concentration, ice water content, and effective mean diameter of pristine ice and snow aggregates with uncertainty estimates.

Evaluations against synthetic observations show that the overall retrieval biases in the combined total microphysical properties are within 5%, and that the errors with respect to the truth are well within the retrieval uncertainty. The partitioning between pristine ice and snow aggregates also agrees well with the truth. Additional evaluations against in-situ cloud probe measurements from a recent campaign for a stratiform cloud system are promising. Our median retrievals have a bias of 98% in total ice number concentration and 44% in total ice water content. This performance is generally better than

the retrieval from empirical relationships. The ability to separate signals of different ice species and to provide their quantitative microphysical properties will open many research opportunities, such as secondary ice production studies and model evaluations for ice microphysical processes.



# 1 Introduction

Ice-containing clouds play an important role in Earth's radiation budget and global precipitation (Baran, 2009; Field and Heymsfield, 2015; Mulmenstadt et al., 2015; Li et al., 2014). Their formation and evolution involve processes of ice nucleation, ice multiplication, aggregation, and riming, which are closely linked to atmospheric conditions and dynamics (DeMott et al., 2011; Gultepe et al., 2017; Field et al., 2017; Korolev et al., 2020). Such complex interactions make it challenging to complete our understanding of these ice microphysical processes and represent them well in models (Korolev
et al., 2017; Morrison et al., 2020).

Polarimetric radar measurements contain information on ice properties and have been proven useful for studying ice microphysical processes (e.g., Kennedy and Rutledge, 2011; Grazioli et al. 2015; Moisseev et al., 2015). Many empirical relationships were developed to provide important bulk properties such as ice water content (e.g., Ryzhkov et al., 1998; Lu et al., 2015), median volume diameter and number concentration (e.g., Murphy et al., 2020), but they cannot inform the
partitioning between ice species. The partitioning is of particular importance for pristine ice crystals because it not only influences snow aggregation rates (Hobbs et al., 1974; Barrett et al., 2019), which impact precipitation production and cloud lifetime (Schmitt and Heymsfield, 2014), but also acts as a major control on cloud phase partitioning (Fusina et al., 2007; Matus and L'Ecuyer, 2017).

However, separating signals of pristine ice from aggregates in polarimetric radar data is challenging, because larger
snow aggregates often dominate the radar reflectivity and mask contributions of smaller pristine ice crystals (Hogan et al., 2002; Keat and Westbrook, 2017). As a result, information from horizontal reflectivity ($Z_H$) alone is insufficient to characterize mixtures of ice hydrometeors (Oue et al., 2018), and it is necessary to incorporate other radar observables in retrieval methods. By exploiting distinct fall behaviours between pristine ice and aggregates, Spek et al. (2008) used $Z_H$, differential reflectivity ($Z_{DR}$) and Doppler spectrum to retrieve particle size distribution (PSD) parameters of pristine ice and
snow aggregates. Without the use of Doppler spectrum, Schrom et al. (2016) used $Z_H$, $Z_{DR}$, and specific differential phase shift ($K_{DP}$) to estimate the PSD of pristine ice in the dendritic growth zone of Colorado winter storms. $K_{DP}$ is a great addition in their approach, since it is mainly determined by ice number concentration. Unfortunately, these three radar observables remain insufficient, and their partitioning between pristine ice and aggregates was weakly constrained. To improve the partitioning, Keat and Westbrook (2017) showed that the relative radar signal contributions of pristine ice embedded in snow
aggregate populations can be quantified using $Z_H$, $Z_{DR}$, and copolar correlation coefficient ($\rho_{hv}$), but they have not attempted to use their partitioning to provide quantitative retrievals of pristine ice number concentration, water content and particle size.

The objective of the paper is to present an ensemble cloud retrieval method (dubbed ENCORE-ice) for simultaneously retrieving the number concentrations, sizes and ice water contents of concurrent pristine ice and snow aggregates from
measurements of $Z_H$, $Z_{DR}$, $K_{DP}$ and $\rho_{hv}$. This framework provides full error statistics and characterizes sub species from radar signals, which is an advance to the existing methods. The polarimetric radar observations and the retrieval method are



detailed in Section 2. The ancillary data sets for evaluations are introduced in Section 3. Section 4 presents evaluation results using synthetic datasets and actual observations from Chilbolton, United Kingdom in 2018. Finally, section 5 summarises the key finding and discusses potential applications.

## 2   Radar observations and ENCORE-ice

### 2.1   Polarimetric radar data

Our retrieval method uses four polarimetric observables. The first observable is the horizontal reflectivity $Z_H$, which provides information on particle size and concentration, but its dependence on size is much stronger. As such, $Z_H$ is dominated by contributions from snow aggregates because their sizes, and thus their backscatter cross-sections, are typically much larger than those of pristine ice crystals. The second observable is the differential reflectivity $Z_{DR}$, which provides information on particle shape and orientation. A $Z_{DR}$ of 0 dB indicates spherical particles because of equal backscattered power in each polarization. Snow aggregates yield low $Z_{DR}$ (about 0–0.6 dB) due to their spheroidal morphology. In contrast, pristine ice particles can yield $Z_{DR}$ of several dB because of their aspect ratios and preferential horizontal orientation when falling. Heterogenous regions with concurrent pristine ice and snow aggregates are therefore associated with higher $Z_{DR}$ than if only snow aggregates were present. The third observable is the co-polar correlation coefficient $\rho_{hv}$, the correlation coefficient between horizontally and vertically backscattered power, which provides information on the diversity of particle shape in a radar sample volume (Kumjian, 2013; Keat et al., 2016). $\rho_{hv}$ is unity in homogenous regions but tends towards lower values (e.g., ~0.97) in the presence of heterogenous hydrometeor types. Finally, the fourth observable is the specific differential phase shift $K_{DP}$, which provides information on particle number concentration, shape, and orientation. Compared to the first two observables that have been widely used in many remote sensing applications, the measurements of $\rho_{hv}$ and $K_{DP}$ are more advanced and their applications remain to be explored more widely.

Our case study is based on polarimetric radar data from the Parameterizing Ice Clouds using Airborne obServationS and triple-frequency dOppler radar data (PICASSO) field campaign in Chilbolton, UK in 2018–2019. During the campaign, the National Centre for Atmospheric Science mobile X-band dual-polarization Doppler weather radar (NXPol; Neely III et al., 2018) operated with 0.98° beam width, 150 m range resolution, and a maximum range of 150 km. The radar performed two back-to-back, fixed-azimuth range-height indicator (RHI) scans every 7 mins, and each scan completed in 18 s. Throughout February 13, 2018, RHI scans were performed along the 243° radial and intercepted by the NCAS-managed Facility for Airborne Atmospheric Measurements (FAAM) aircraft on several occasions, providing a unique opportunity for evaluation. Key characteristics of NXPol are summarized in Table 1.



**Table 1.** Characteristics of the NXPol polarimetric radar. Further specifications and details can be found in Neely III et al. (2018).

| Parameter | NXPol |
| --- | --- |
| Center wavelength (mm) | 31.98 |
| Transmit/receive polarization | H+V/H+V |
| Beamwidth (°) | 0.98 |
| Pulse width (μs) | 1 |
| Scan rate (° s$^{-1}$) | 5 |
| Sensitivity (dBZ) | −11 (at 100 km) |
| Maximum range (km) | 150 |
| Gate Resolution (m) | 150 |

## 2.2 ENCORE-ice

ENCORE is an ensemble-based retrieval method that has previously been used to retrieve three-dimensional cloud microphysical properties (Fielding et al., 2014) and one-dimensional cloud and drizzle properties (Fielding et al., 2015), but several key components are modified here for ice retrieval.

### 2.2.1 Particle size distribution

We approximate the PSD of pristine ice and aggregates by normalized Gamma distributions, given as (Testud et al., 2001):

$$n(D) = N_0 f_\mu(D), \tag{1}$$

where $N_0$ is the normalized number concentration, and $n$ is the number concentration at a given maximum particle dimensions $D$. The choice of the size descriptor in equation (1) is because in-situ cloud probe data and the ice scattering database are both given based on the maximum particle dimension. The function $f_\mu$ is defined as:

$$f_\mu(D) = \frac{6}{3.67^4} \cdot \frac{(3.67+\mu)^{4+\mu}}{\Gamma(4+\mu)} \left(\frac{D}{D_0}\right)^\mu \cdot \exp\left[-(3.67+\mu)\frac{D}{D_0}\right], \tag{2}$$

where $\mu$ is the shape parameter of the PSD, and $D_0$ is the diameter used for normalizing $D$. Following Mason et al. (2018), we assume a constant shape parameter of $\mu = 2$. Several studies have shown that the retrieved ice water content is relatively insensitive to the choice of shape parameter (e.g., Delanoë et al., 2005; Spek et al., 2008); we also found that our number concentration retrieval is not sensitive to $\mu$ either.

From the PSD, the total ice number concentration ($N_I$) and the total ice water content ($q_I$) can be respectively computed
by:

$$N_I = \int_0^\infty n(D)dD = N_P + N_A, \text{ and} \tag{3}$$





$$q_I = \int_0^\infty m(D)n(D)dD = q_P + q_A,$$ (4)

where the subscripts $P$ and $A$ denote contributions from pristine ice and snow aggregates, respectively, and $m(D)$ is the mass at a given maximum particle dimensions $D$. The mass-size relationship can be formulated as:

$$m(D) = aD^b,$$ (5)

where $a$ and $b$ are the pre-factor and exponent, respectively. These coefficients depend on ice habit and have been estimated from past aircraft in-situ and surface observation as shown in Table 2. From the PSD, we also define and calculate the effective mean diameters ($D_{eff}$) as:

$$D_{eff} = \frac{\int_0^\infty n(D)D^4 dD}{\int_0^\infty n(D)D^3 dD},$$ (6)

which is the ratio of the 4th to the 3rd moment of PSD. To compare our retrieval with the empirical estimates (as discussed in Section 3), we also calculate an effective mean diameter using the equivalent melted diameter ($D_{mlt}$) as the size descriptor, defined as:

$$D_{eff,mlt} = \frac{\int_0^\infty n(D_{mlt})D_{mlt}^4 dD_{mlt}}{\int_0^\infty n(D_{mlt})D_{mlt}^3 dD_{mlt}},$$ (7)

where

$$D_{mlt} = \left[\frac{6m(D)}{\pi\rho_w}\right]^{\frac{1}{3}} = \left[\frac{6aD^b}{\pi\rho_w}\right]^{\frac{1}{3}}, \text{ and}$$ (8)

$\rho_w$ is water density.

**Table 2.** Examples of mass-size relationships (taken from Mason et al., 2018).

| Habit | $a$ (g cm$^{-b}$) | $b$ | Reference |
|---|---|---|---|
| Stellar | 0.00027 | 1.67 | Mitchell (1996) |
| Hexagonal columns | 0.000907 | 1.74 | |
| Broad branches | 0.000516 | 1.80 | |
| Sector-like branches | 0.00142 | 2.02 | |
| Bullet rosettes | 0.00308 | 2.26 | |
| Side planes | 0.00419 | 2.3 | |
| Hexagonal plates | 0.00739 | 2.45 | |
| Aggregates | 0.0028 | 2.1 | |
| Aggregates | 0.0039 | 1.9 | Szyrmer and Zawadzki (2010) |
| Unrimed dendrites | 0.001263 | 1.912 | Erfani and Mitchell (2017) |
| Mixed (large-scale and convectively generated ice clouds) | 0.007 | 2.2 | Heymsfield et al. (2010) |



### 2.2.2 The basis of ENCORE-ice

The state vector ($\boldsymbol{x}$, i.e., variables to be retrieved) for each ensemble member is defined as:

$$\boldsymbol{x} = \left( \log_{10} N_{0,\mathrm{P}}^{(i=1\ldots G)}, \log_{10} D_{0,\mathrm{P}}^{(i=1\ldots G)}, \log_{10} N_{0,\mathrm{A}}^{(i=1\ldots G)}, \log_{10} D_{0,\mathrm{A}}^{(i=1\ldots G)} \right), \tag{9}$$

where the superscript $i$ represents the index of the range gate, and the total number of gates to be retrieved is $G$. Let us use $Q$ members to form an ensemble, i.e.,

$$\mathbf{X} = \{\boldsymbol{x}_1, \ldots, \boldsymbol{x}_Q\} \tag{10}$$

such that the mean of $\mathbf{X}$ represents the best estimate of the state vector, and the spread of the ensemble members around the mean represents the uncertainty in the best estimate.

Using the Iterative Stochastic Ensemble Kalman Filter approach (Evensen et al., 2019), each ensemble member is updated based on:

$$\boldsymbol{x}_k^a = \boldsymbol{x}_k^f + \boldsymbol{E}_k^f \mathbf{w}_k, \tag{11}$$

in which $\boldsymbol{x}_k^f$ and $\boldsymbol{x}_k^a$ are the prior and posterior ensemble member $k$, respectively, and

$$\boldsymbol{E}_k^f = \left[ \boldsymbol{x}_1^f - \overline{\mathbf{X}}^f, \ldots, \boldsymbol{x}_Q^f - \overline{\mathbf{X}}^f \right] \tag{12}$$

is the initial ensemble matrix with the prior mean ($\overline{\mathbf{X}}^f$) subtracted, and $\mathbf{w}_k$ are weight vectors that are calculated from iteratively minimizing the following cost function:

$$J(\mathbf{w}_k) = \frac{1}{2} \mathbf{w}_k^T \mathbf{w}_k + \frac{1}{2} \left( \boldsymbol{y} - \boldsymbol{h}(\boldsymbol{x}_k^f + \boldsymbol{E}_k^f \mathbf{w}_k) - \varepsilon_k \right)^T \boldsymbol{R}^{-1} \left( \boldsymbol{y} - \boldsymbol{h}(\boldsymbol{x}_k^f + \boldsymbol{E}_k^f \mathbf{w}_k) - \varepsilon_k \right). \tag{13}$$

In equation (13), the observation vector $\boldsymbol{y}$ is defined as gate-by-gate radar observables:

$$\boldsymbol{y} = \left( Z_{\mathrm{H}}^{(i=1,\ldots,G)}, Z_{\mathrm{DR}}^{(i=1,\ldots,G)}, -\ln K_{\mathrm{DP}}^{(i=1,\ldots,G)}, -\ln \rho_{\mathrm{HV}}^{(i=1,\ldots,G)} \right), \tag{14}$$

where $Z_{\mathrm{H}}$ and $Z_{\mathrm{DR}}$ are in dB. $\boldsymbol{h}(\boldsymbol{x})$ represents the forward model for simulating polarimetric radar observables from the state vector $\boldsymbol{x}$, and $\varepsilon_k$ is a random perturbation vector drawn from the observation error distribution, which is estimated to be Gaussian with mean zero and covariance matrix $\boldsymbol{R}$ (Evensen et al., 2019, with modification from Van Leeuwen, 2020). The covariance matrix $\boldsymbol{R}$ is diagonal with standard deviations given in Table 3.





**Table 3.** Estimated observational errors for X-band observables based on standard and benchmark procedures, adapted from Bringi and Chandrasekar (2004; pp 359–376) and Wang and Chandrasekar (2009).

| Observable | Description | Uncertainty |
|:---:|---|---|
| $Z_H$ | Horizontal reflectivity | 0.5 dBZ |
| $Z_{DR}$ | Differential reflectivity | 0.05 dB |
| $K_{DP}$ | Specific differential phase shift | *10% |
| $\rho_{HV}$ | Co-polar correlation coefficient | #1 % |

* estimated by the uncertainty of 0.05 ° km$^{-1}$ for a typical value of $K_{DP} = 0.5$ ° km$^{-1}$.

165     # estimated by the uncertainty of 0.01 for $\rho_{HV} = 0.95$.

### 2.2.3   Simulating radar observables for $h(x)$

To model polarimetric radar observables from the assumed PSD, knowledge of the single scattering properties of ice particles is required. Many scattering databases of realistically shaped ice particles at radar wavelengths are available and we used Lu et al. (2016) because of the following considerations. Several existing scattering databases assume total random orientation of the scatterers, e.g., Liu (2008), Hong et al. (2009), Kuo (2016) and Eriksson et al. (2018). Such assumption cannot explain polarimetric radar signals which are produced by non-spherical scatterers with preferred orientations with respect to the zenith direction. The database of Brath et al. (2020) assumes scatterers possess arbitrary fixed orientations relative to the zenith direction, but only includes hexagonal plates and aggregates consisting of hexagonal plates. We found that the database described in Lu et al. (2016) fits our needs in the current polarimetric radar study, since it contains all necessary polarimetric scattering data in many fixed orientations of a large variety of ice crystal species, including plates, columns, dendrites, and aggregates. The single scattering properties for each species are available for a range of crystal maximum dimensions, thickness ratios, and types. The pristine habits generally begin at ~0.1 mm and do not exceed 6 mm, whereas the aggregates begin at ~0.4 mm and extend to 18–45 mm approximately. Multiple morphological realizations per maximum dimension are available for dendrites and aggregates to account for their complexities.

The scattering calculations were conducted using the generalized multi-particle Mie method (GMM; Xu, 1995) and the discrete dipole approximation (DDA; Yurkin and Hoekstra, 2011). We used properties calculated from GMM, because DDA calculations are not available for aggregates. Specifically, we use the amplitude scattering matrix elements in the forward and backward direction for horizontally and vertically polarized radiation, denoted as $S_{hh}^{f,b}$ and $S_{vv}^{f,b}$ where the superscript and subscript respectively represent the scattering direction (i.e., forward or background) and the polarization status (horizontally or vertically). From the assumed PSD and the amplitude scattering matrix elements, radar observables for a single sample volume containing multiple ice particle habits can be derived as shown in Appendix A.



## 2.3 Practical considerations

There are several practical considerations for ENCORE-ice implementation. The first consideration is ice habit. The scattering database provides three habits (plates, dendrites, and columns) for pristine ice. Since the temperature found in PICASSO mostly ranged between –5°C and –25°C, all three types of pristine ice can be the preferred habit, as shown in Fig. 1. Currently, we ran our retrieval algorithm for all three habits, and then selected the most appropriate one based on the agreement in the measured and forward simulated radar observables. Similarly, the scattering database provides five types of

aggregates; two of them were constructed using ice columns (LD-N1e and HD-N1e), three of them using stellar ice crystals (LD-P1d, LDt-P1d and HD-P1d). The aspect ratios of aggregates in nature are known to vary depending on the pristine habits that comprise them. Defining an aspect ratio as the ratio of the lengths of the minor axes to the major axes, Garrett et al. (2015) and Jiang et al. (2017) observed aspect ratios ranging from 0.3 to 0.6 for falling aggregates at the surface. As shown in Fig. 2, among the aggregates available in the scattering database, LDt-P1d and HD-P1d exhibit aspect ratios within

the observed range. The mass-size relationship for LDt-P1d is based on ordinary dendritic crystal with coefficient $a$ of 0.000482 and coefficient $b$ of 1.97 (Kajikawa, 1989; Botta et al., 2011) in units of cgs same as Table 2, whereas the mass-size relationship for HD-P1d is based on aggregates of thin plate with coefficient $a$ of 0.00145 and coefficient $b$ of 1.80 in units of cgs (Mitchell and Heymsfield, 2005; Botta et al., 2011). The mass-size relationship of HD-P1d is very close to unrimed aggregates (Erfani and Mitchell, 2017) and more aligned to values in the recent literature listed in Table 2. Hence,

we select HD-P1d as the prescribed choice for aggregates.

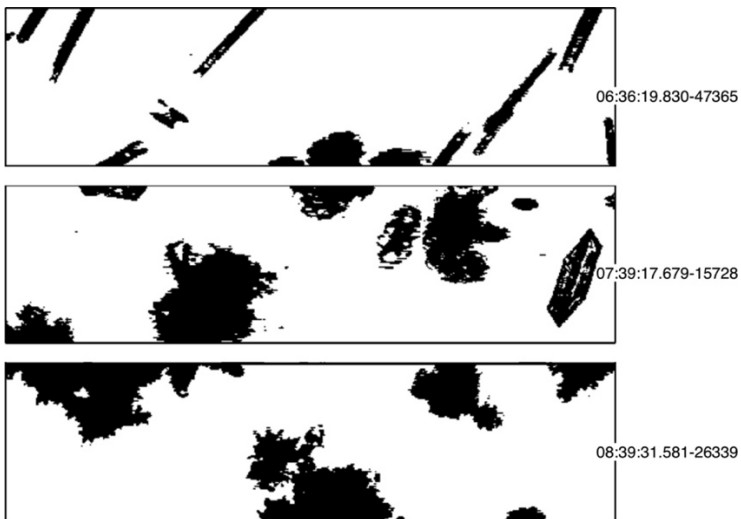

**Figure 1.** Examples of particle images from the Stratton Park Engineering Company Two-Dimension Stereo (2DS) probe, showing the presence of (a) column, (b) plate and (c) dendrite on 13 February 2018. Each image frame is 1.28 mm high, taken from one of the probe
channels only since the other channel was not working properly on this day.



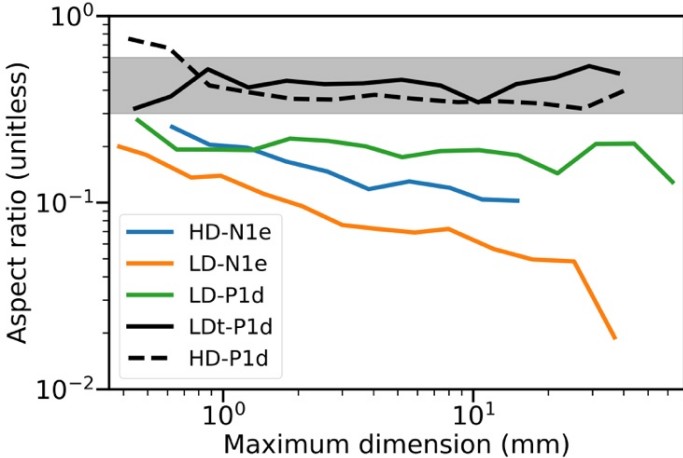

**Figure 2.** Aspect ratios of various aggregate types available in the scattering database as a function of their maximum dimensions. Aspect ratio is defined as the ratio of the sizes of the minor axes to the major axes. The grey shading between 0.3 and 0.6 represents the typical range of snow aggregate aspect ratios observed in nature (Garrett et al. 2015; Jiang et al. 2017).

The second consideration is the prior used to generate the first guess for ensemble members. Using over 70 hours of in-situ aircraft observations from a wide range of field campaigns spanning diverse cloud and temperature regimes, Delanoë et al. (2014) characterized the PSDs of ice particles by the normalized Gamma distribution. They found that $N_0$ ranged between 1 $L^{-1}$ $mm^{-1}$ and 10,000 $L^{-1}$ $mm^{-1}$ with a mean 100 $L^{-1}$ $mm^{-1}$, and the median volume diameter (MVD) ranged between 0.2–0.8 mm with a mean of 0.5 mm for the temperature zone of –10°C to –20°C. Additionally, Tiira et al. (2016) analysed surface measurements of ice particle number concentration from the Precipitation Imaging Package during the Biogenic Aerosols – Effects on Clouds and Climate field campaign. They found that $N_0$ ranged mainly from 1 $L^{-1}$ $mm^{-1}$ to 100 $L^{-1}$ $mm^{-1}$ and MVD ranged from 0.5 mm to 5 mm. As these were surface based, the measured PSDs from Tiira et al. (2016) are more representative of the characteristics of snow aggregates. Note that these values were derived using the equivalent melted diameter as the size descriptor, not the maximum particle dimension. Hence, these values are used only to point out a possible range and serve as a starting point for us to construct the prior.

Based on these observational ranges mentioned above, our prior is designed as follows. We started with the lowest radar gate, randomly assigning $(N_{0,P}, D_{0,P}, N_{0,A}, D_{0,A})$ from normal distributions with the means and standard deviations listed in Table 4. Next, we applied a slope for each ray to provide initial guesses for other radar gates. The slope was randomly selected from a normal distribution described in Table 4. Among $N_{0,P}, D_{0,P}, N_{0,A},$ and $D_{0,A}$, $N_{0,P}$ likely increases with height, because the prevalence of active ice nuclei is a function of temperature and thus a function of height as well (DeMott et al., 2010); however, the rest lack a clear dependence on height (e.g., Field et al., 2005). Hence, the slopes applied for $N_{0,P}$ in the prior are assumed to have a positive mean, while the slopes applied for $D_{0,P}, N_{0,A},$ and $D_{0,A}$ have a slightly negative mean. The small negative slope means are necessary to cover an appropriate range of state variables for radar gates at higher



altitudes. Finally, red (AR1) noise was added over the vertical with a correlation coefficient of 0.999 and a zero-mean random perturbation with a standard deviation that is half that of the lowest radar gate.

Compared to values reported in Delanoë et al. (2014) and Tiira et al. (2016), we have chosen lower means for $N_{0,P}$ and $N_{0,A}$ to start with. This is because the state vector space is the logarithm of $N_0$. Positive slopes make the changes of $N_0$ much

more dramatic in the vertical than those with negative slopes. As a result, large starting values of $N_{0,P}$ and $N_{0,A}$ will lead to unrealistic high concentrations at higher altitudes in the prior. In contrast, we choose larger means for $D_{0,P}$ and $D_{0,A}$, because of the assumed negative slopes in both $D_{0,P}$ and $D_{0,A}$. In general, the range in our prior is large, approximately two orders of magnitude in state vector variables. For such a wide-spread prior the solution will be dominated by the observations.

**Table 4.** The prior and uncertainty used in ENCORE-ice. The means at lowest radar gate are given in the physical state space, and the rest are in the transformed state space (i.e., $\log_{10}$). Retrieval is performed using two different sets of the prior; the second set uses values in the parenthesis and the rest remain unchanged. All radar gates above the lowest gates are perturbed by an AR1 red noise process with a vertical correlation of 0.999 and a standard deviation that is half of the standard deviation at lowest gates.

| Variable | Pristine Ice | | Aggregate | |
|---|---|---|---|---|
| | $N_{0,P}$ | $D_{0,P}$ | $N_{0,A}$ | $D_{0,A}$ |
| *Value at lowest radar gate* | | | | |
| Mean | 50 (or 5) L$^{-1}$ mm$^{-1}$ | 1 mm | 5 L$^{-1}$ mm$^{-1}$ | 4 (or 1) mm |
| Standard deviation | 0.15 | 0.3 | 0.15 | 0.3 |
| *Slope in the vertical* | | | | |
| Mean (km$^{-1}$) | 1 | −0.5 | −0.5 | −0.5 |
| Standard deviation (km$^{-1}$) | 0.2 | 0.02 | 0.2 | 0.02 |


The third consideration is the number of the ensemble members used in the ENCORE-ice. Ideally, a large ensemble size is needed to ensure that the sampled prior is representative and so is the solution. However, a large ensemble size is computationally expensive. Therefore, we applied a localization scheme to reduce the required number of ensemble members so that we shorten the computational time while achieving the same mean retrieval and associated uncertainty. The

localization scheme operates on each gate and takes only observations close to that gate into account to find the solution. This is implemented by multiplying the observation error variance of each observation with an exponential function of the distance between that observation and the gate that is being updated, such that observations far from the gate have less influence. The influence radii vary linearly with height, one gate at the lower level and about five gates at the upper level. Using our synthetic datasets, we have found that 50 ensemble members with the localization scheme is able to produce

similar mean retrievals and associated uncertainty as a non-localized ensemble of size 500. The number of iterations is set to 20, although the solutions often have converged at the 10th iteration.

Finally, all radar data underwent the following quality checks and corrections before being used for retrieval:





- $Z_H$ and $Z_{DR}$ were corrected for attenuation due to liquid water, using the method described in Bringi and Chandrasekar (2001, Page 490–512). The attenuation due to ice at X-band is negligible and thus ignored here (Vivekanandan et al., 1999).

- Systematic biases in $Z_{DR}$ were identified using zenith-pointing $Z_{DR}$ observations. As hydrometeors produce $Z_{DR}$ of 0 dB when viewed at zenith due to their spherical symmetry (e.g., for raindrops) or lack of preferential azimuthal orientation (e.g., for ice particles), any residual $Z_{DR}$ can be treated as bias and removed (Seliga et al., 1981). We have found the $Z_{DR}$ correction factors to be 0.2 dB for the PICASSO cases.

- $K_{DP}$ is calculated using the method of Wang and Chandrasekar (2009).

- Once all corrections are applied, measurement noises were removed using a cubic spline approach (Craven and Wahba, 1979).

Additionally, to ensure that gates are associated with sufficient information for our method, we exclude gates that exhibit one or more of the following:

- Gates within 500 m of the 0°C level, avoiding contamination from liquid hydrometeors in the radar sample volume, because our state vector is not designed for that.

- Gates where $\rho_{HV}$ exceeds 1.0, because these values are unphysical.

- Gates with a signal-to-noise ratio (SNR) less than 20 dB, because of a lack of detectable hydrometeors. This threshold is chosen because the precipitating region and its surrounding area typically have SNR values larger than 30–40 dB.

- Radar rays with elevation angles greater than 50°, because $Z_{DR}$ tends towards 0 dB at higher elevation angles and polarimetric information becomes ambiguous.

- Gates with $Z_{DR}$ below 0.25 dB, regardless of their elevation angles, because the relative contributions of pristine ice and aggregates become ambiguous at lower values, as indicated in Keat and Westbrook (2015).

- Gates with $K_{DP}$ below 0.1 ° km$^{-1}$ to ensure a sufficient number concentration of pristine ice. Negative $K_{DP}$, indicative of conical graupel (Aydin and Seliga 1984) or the vertical reorientation of pristine ice crystals in the presence of thunderstorm electric fields (Hubbert et al. 2014). Since the state vector only includes pristine ice and aggregates, we exclude such gates as well.

## 3   Independent observations and retrievals for evaluations

### 3.1   In-situ aircraft measurements from PICASSO

During PICASSO, the FAAM aircraft performed multiple transects from Chilbolton to Dorset (50.82 °N, 2.56 °W) at varied altitudes. Figure 3 depicts the flight path for 13 February 2018, which was a typical pattern during the campaign.





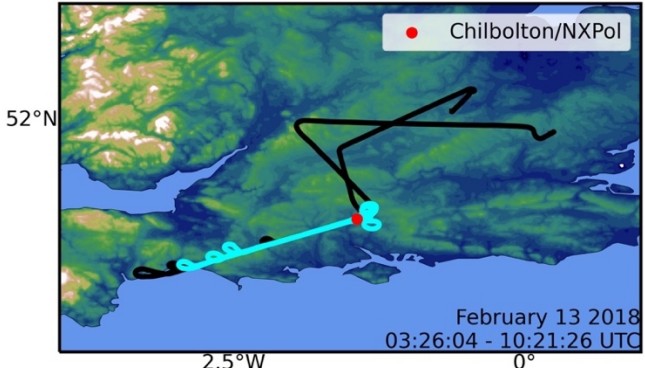

**Figure 3.** Flight paths on 13 February 2018 between 3:26 UTC and 10:21 UTC. The red dot denotes the location of NXPol in Chilbolton, UK, while the path in cyan denotes the path during 6–9 UTC in which retrievals are evaluated in Section 4.2.

To evaluate our cloud retrieval, we use in-situ measurements of liquid water content and total water content (i.e., the sum of ice and liquid water contents) from a Nevzorov probe, and PSD measurements from a High-Volume Precipitation Spectrometer (HVPS, SPEC Inc, USA). The HVPS is an optical array particle imaging probe, which collects images of ice crystals with a pixel resolution of 150 µm. Size distributions of particles between 75 and 19275 µm were derived from their images; see Crosier et al. (2011) for a description of the data processing and quality control procedures used, and O'Shea et al. (2021) for discussions of the sources of uncertainties. All in-situ datasets were averaged to 5 s intervals for statistical reliability (Protat et al., 2007). We only use in-cloud samples, defined as having an ice water content (IWC) greater than 0.01 g m$^{-3}$. Additionally, although our retrieval provides microphysical properties of pristine ice and aggregates separately, we focus on evaluating bulk properties to avoid the ambiguity introduced by applying a threshold to separate these two species in observed PSDs.

Three bulk properties are used for evaluations. Firstly, the total ice number concentration (denoted as $N_{\text{I,HVPS}}$) is calculated by integrating the observed PSD. The associated counting uncertainty is estimated as:

$$\frac{100\%}{\sqrt{N_{\text{I,HVPS}} \cdot V_{\text{HVPS}} \cdot \Delta t}}, \tag{15}$$

where $\Delta t$ is the HVPS sampling time resolution, and $V_{\text{HVPS}}$ is the sample volume, approximately 310 L s$^{-1}$. Secondly, IWC, denoted as $q_{\text{I,NEV}}$, was derived by taking the difference between the total and liquid water contents measured by the Nevzorov probe. Similar to Abel et al. (2014), both the total and liquid water contents were corrected for changes in aircraft altitude and environmental conditions. Finally, effective mean diameters from HVPS PSDs, $D_{\text{eff,HVPS}}$, defined as:

$$D_{\text{eff,HVPS}} = \frac{\int_0^\infty n_{\text{HVPS}}(D)D^4 dD}{\int_0^\infty n_{\text{HVPS}}(D)D^3 dD}. \tag{16}$$

were calculated, using the same definition as equation (6).



These simultaneous evaluations allow us to indirectly examine whether the partitioning between pristine ice and aggregates is appropriate. A more direct comparison would be ideal but requires classifying each individual particle in image data, which is not trivial and beyond the scope of this work.

## 3.2 Bulk ice properties from empirical relationships

As mentioned in Sec. 1, several studies have proposed empirical relationships for estimating IWC, particle size, and ice number concentration. In this study, we compare our retrieval with estimates from Ryzhkov and Zrnic (2019), because of their availability of the ice number concentration estimates. The relationships in Ryzhkov and Zrnic (2019) were based on theoretical calculations, using an assumed exponential size distribution for twelve ice habits. Their method takes advantage of the features that the reflectivity difference between horizontal and vertical polarization ($Z_{DP}$) is proportional to the third

moment of PSD and that $K_{DP}$ is proportional to the first moment of PSD. As a result, the ratio of $Z_{DP}$ to $K_{DP}$ is proportional to the second moment of the PSD and can be used to estimate the mean volume diameter of ice particles (Murphy et al. (2020):

$$D_{emp} = -0.1 + 2\left(\frac{Z_{DP}}{K_{DP}\lambda}\right)^{\frac{1}{2}}. \tag{17}$$

where $D_{emp}$ is the mean volume diameter in mm with the subscript *emp* denoting empirical estimates; $Z_{DP}$ is in units of mm⁶

m⁻³; and $K_{DP}$ is in ° km⁻¹. Murphy et al. (2020) also estimated the number concentration and IWC using:

$$\log_{10} N_{emp} = 0.1Z_H - 2\log_{10}\frac{Z_{DP}}{K_{DP}\lambda} - 1.11; \tag{18}$$

$$q_{I,emp} = 0.004\left(\frac{K_{DP}\lambda}{1-Z_{DR}^{-1}}\right). \tag{19}$$

where $N_{emp}$ and $q_{I,emp}$ are in units of L⁻¹ and g m⁻³, respectively; $Z_H$ is in units of dBZ; $Z_{DR}$ is unitless, and $\lambda$ is the radar wavelength in mm. For convenience, we refer to retrievals from these empirical relationships as "Murphy20" hereafter.

Based on the evaluation conducted by Murphy et al. (2020) for a stratiform region of a mesoscale convective system over Oklahoma, $N_{emp}$ scatters significantly with respect to in-situ measurements; $q_{I,emp}$ and $D_{emp}$ tends to be systematically biased low, but outperformed other empirical relationships. Since, in theory, these derived relationships are not sensitive to ice particle shape and orientation, they remain a good starting point for intercomparisons.

Note that these empirical relationships are designed for radar volumes that only include one species. Hence, if a radar

volume is known to include a mixture of different species, caution should be exercised when interpretating their results. Additionally, equation (17) was derived using equivalent volume diameter as the size descriptor in PSD, and thus $D_{emp}$ cannot be used directly for comparisons to our retrieval that is based on maximum particle dimension as the size descriptor. Instead, we need to trace back their derivations to find their retrieved PSD, convert the equivalent volume diameter to the


equivalent melted diameter ($D_{\mathrm{mlt}}$), and then calculate the effective mean diameter ($D_{\mathrm{eff,mlt}}$) using equation (7) for
intercomparisons. The details can be found in Appendix B.

## 4   Results

### 4.1    Evaluation using synthetic data

In this section we use synthetic polarimetric radar data to evaluate our retrieval and identify any potential issues. The
synthetic dataset was generated as follows. We first generated 501 profiles from the prior used in the ENCORE-ice, and then
randomly selected a profile that has a relatively wide range of $Z_{\mathrm{H}}$ and $Z_{\mathrm{DR}}$ for testing. Along with the forward model
described in Section 2.2.3, this selected profile is used to generate synthetic radar measurements and serves as the "truth" in
this evaluation experiment. Because the truth profile and the initial ensemble members were generated from the same prior
and used the exact same forward models, any retrieval error found in this experiment is due to the combination of the
observed uncertainty and the retrieval method itself only. Hence, the design of this experiment does not allow us to evaluate
errors due to the representativeness of forward models or the prior, which likely exist in real world applications.

Figure 4 shows the synthetic radar measurements over 20 gates with a resolution of 50 m at a given elevation angle of
30°, based on the truth profile shown in Fig. 5. The chosen number of gates is arbitrary but represents a frequent scenario in
the radar scans collocated with in-situ data during PICASSO. In this scenario, the total ice number concentration is
dominated by pristine ice, and the total ice water content is dominated by aggerates. The truth has a $N_{\mathrm{P}}$ range between 5–20
L$^{-1}$, and a $N_{\mathrm{A}}$ range between 1–3 L$^{-1}$; $D_{0,\mathrm{P}}$ ranges between 0.4–1 mm, while $D_{0,\mathrm{A}}$ ranges ~2–5 mm. The combined $D_{\mathrm{eff}}$ varies
from 1.5 mm to ~5 mm, generally close to $D_{0,\mathrm{A}}$ as expected, since it is weighted by size to the third power and mainly
controlled by the species with large particle sizes.

The forward-modelled observables of the ENCORE-ice solution in Fig. 4 agree well within the uncertainty of the
synthetic values, providing confidence in retrievals. As shown in Fig. 5, the retrieval captures the vertical trend of the truth;
the retrieval uncertainty estimated from the spread of the ensemble members also appear reasonable, since the truth falls
within the retrieval uncertainty. Because many combinations of $N_{0,\mathrm{A}}$ and $D_{0,\mathrm{A}}$ could lead to the same $Z_{\mathrm{H}}$, we see some
compensating effects between $N_{0,\mathrm{A}}$ and $D_{0,\mathrm{A}}$ in aggregates at the lower layer. The errors are compensated so that the error in
the total ice water content is not enhanced, as shown in Table 5. Overall, the retrieval biases in the combined total number
concertation, water content, and effective diameter properties are within 5%, and the root-mean-square-errors are small (see
Table 5).

To conclude, this evaluation experiment demonstrates that the combination of these four radar observables is appropriate
and the current observational uncertainty is sufficient for us to separate signals of pristine ice from aggregates. The errors in
retrieved pristine ice properties are small, and thus further physical interpretation based on the associated vertical profiles
can be made to understand the underlying microphysical processes. For aggregates, the errors in retrieved size diameter and





water content are small, but the vertical variations of retrieved number concentration may not follow the truth exactly due to the possible compensating effects between number concentration and particle size.

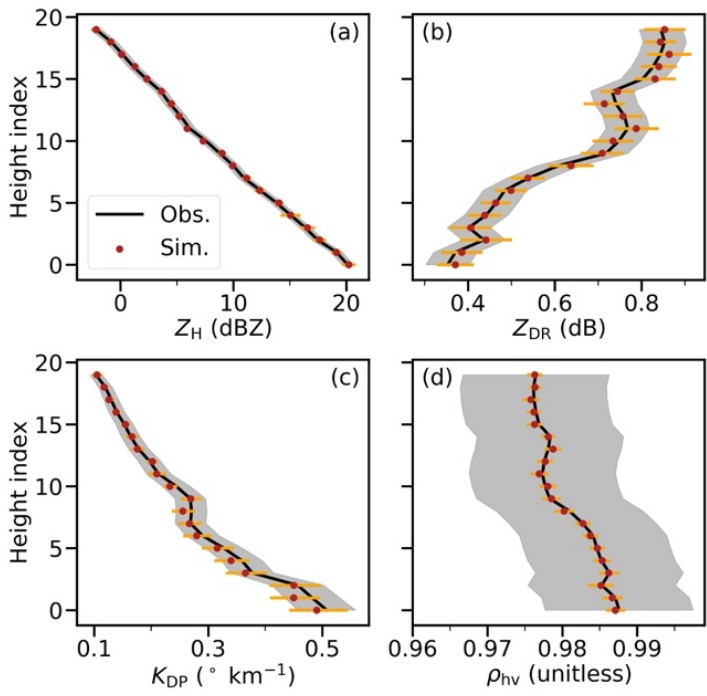

**Figure 4.** Profiles of (a) $Z_H$, (b) $Z_{DR}$, (c) $K_{DP}$ and (d) $\rho_{HV}$ for synthetic observations (black line) calculated from the ice cloud properties
given in Fig. 5, and for the mean of forward simulations from the ensemble (red dots). Grey shading denotes the observational uncertainties given in Table 3, while error bars in orange denote retrieval uncertainty calculated as the one-standard deviation spread of the ensemble simulations.


**Table 5.** Truth means, and the means, root-mean-square-error (RMSE), and biases in retrieval in the synthetic dataset experiment.

| | Ice number concentration $(N_P, N_A; \text{L}^{-1})$ | | Normalization diameter $(D_{0,P}, D_{0,A}; \text{mm})$ | | Total number concentration $(N_I; \text{L}^{-1})$ | Total ice water content $(q_I; \text{g m}^{-3})$ | Combined effective diameter $(D_{\text{eff}}; \text{mm})$ |
|---|---|---|---|---|---|---|---|
| | Pristine ice | Aggregate | Pristine ice | Aggregate | | | |
| True mean | 13.270 | 0.882 | 0.677 | 3.165 | 14.152 | 0.139 | 2.965 |
| Ret. mean | 12.836 | 0.988 | 0.692 | 3.127 | 13.824 | 0.145 | 2.927 |
| RMSE | 0.87 | 0.22 | 0.03 | 0.14 | 0.74 | 0.01 | 0.12 |
| Bias (%) | −3.3 | 12.0 | 2.2 | −1.2 | −2.3 | 4.3 | −1.3 |

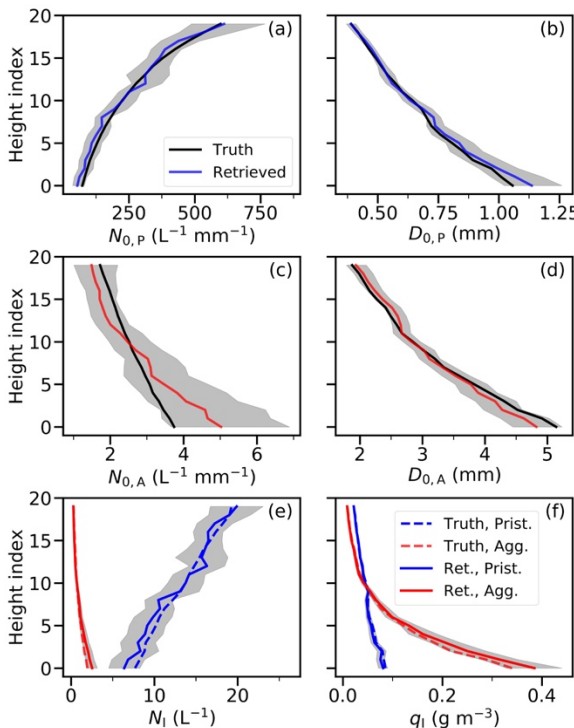

**Figure 5.** Profiles of (a) normalized number concentration and (b) normalization diameter for pristine ice, (c) normalized number concentration and (d) normalization diameter for aggregates. (e) represents the total number concentrations and (f) represents ice water contents of pristine ice and aggregates. Truth is denoted by black solid lines in (a)–(d), and by dashed lines in (e) and (f). The retrieved ensemble means are denoted by solid blue and red lines with shading that represents the one-standard deviation spread of the ensemble members. The habit of pristine ice is plate in this experiment.

## 4.2    Evaluation using PICASSO data

The case of 13 February 2018 from the PICASSO campaign represents a stratiform precipitating cloud system associated with a frontal passage. Using a radar scan at 8:37 UTC as an example, Fig. 6 shows a significant area with reduced $\rho_{HV}$, and enhanced $Z_{DR}$ and $K_{DP}$ at ~3 km height, which suggests the presence of enhanced pristine ice embedded in snow aggregates. Based on the temperatures measured by the aircraft (Fig. 7), this area is in a temperature zone of –15°C, and thus the preferred ice habit is likely to be dendrite and plate for this radar scan. During this radar scan, FAAM was too far away to provide meaningful comparison, but cloud images showed that dendrites were present most of time during this period.

Figure 8 shows detailed retrieval performance for a ray taken from the radar scan in Fig. 6. For this case, retrievals using the dendrite habit perform best; the habit suggested by our retrieval is consistent with observed particle images. As shown in Figs. 8(a)–(d), the forward modelled radar observables agree well with the observed vertical profiles. The normalization diameters for pristine ice and aggregates ($D_{0,P}$, $D_{0,A}$) are about 2.5 mm and 5–6 mm, respectively. Retrieved $N_I$ is relatively





constant at ~5 L$^{-1}$, due to the opposite vertical variations between $N_P$ (increasing with height) and $N_A$ (decreasing with height). In contrast, $q_I$ decreases with height from 1 g m$^{-3}$ to 0.2 g m$^{-3}$, because both $N_A$ and $D_{0,A}$ decrease with height.

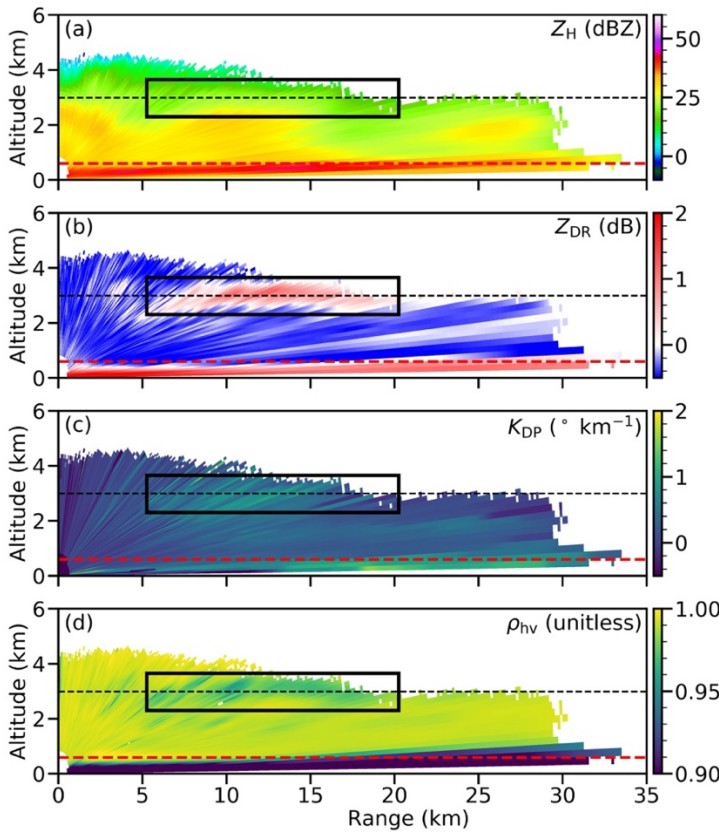


**Figure 6.** Height-range plots of observed (a) $Z_H$, (b) $Z_{DR}$, (c) $K_{DP}$ and (d) $\rho_{HV}$ from the RHI scan at 8:37 UTC on 13 February 2018 during the PICASSO field campaign. The red dashed line denotes the 0°C level, while the black dashed line denotes the approximate flight altitude of FAAM during the scan. The black polygon denotes the region that has enhanced $Z_{DR}$ and $K_{DP}$, and reduced $\rho_{HV}$.

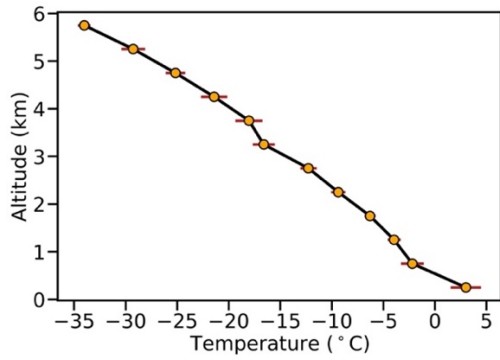


**Figure 7.** A temperature profile composited from aircraft in-situ data during 6–9 UTC on 13 February 2018. Data between 6:33:30 – 6:39:20 UTC were unphysical (see Fig. 9b) and thus excluded. The error bars represent one standard deviation of sampled observations.

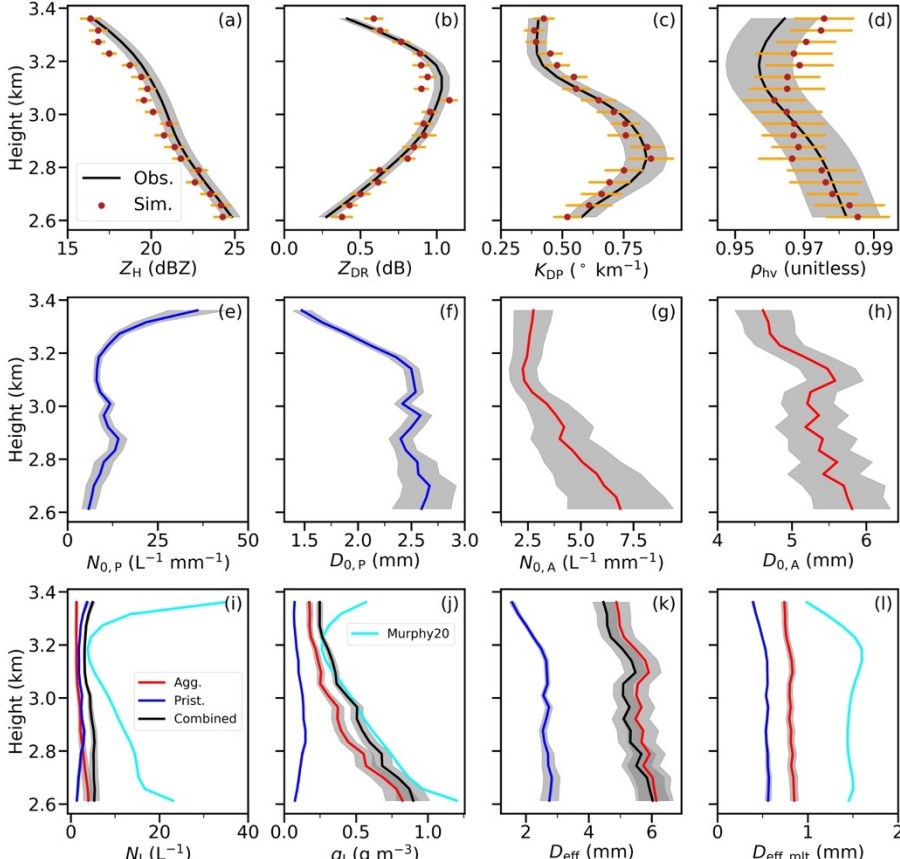

**Figure 8.** Retrieval performance for a radar ray at 8:37:39 UTC. Observed and forward simulated profiles of (a) $Z_H$, (b) $Z_{DR}$, (c) $K_{DP}$ and (d) $\rho_{HV}$. The shading in (a)–(d) represent the observational uncertainty. The red dots represent the mean of the ensemble simulations, and the error bars represent one standard-deviation in forward simulations. (e)–(j) represent the retrieved mean normalized pristine ice number concentration, pristine ice normalization diameter, normalized aggregate number concentration, aggregate normalization diameter, the total number concentrations, and the total ice water content, respectively. (k) represents the individual and combined effective mean diameters
using the maximum particle dimension as the size descriptor, while (l) represents the individual and combined effective mean diameters using the equivalent melted particle as the size descriptor. The shading in (e)–(l) represent one standard deviation uncertainty in retrieval. For comparisons, retrievals from Murphy20 are co-plotted in (i), (j) and (l).

Compared to Murphy20 retrievals, a few findings stand out. Firstly, retrieved $q_I$ profiles from two methods follow each
other closely. This is not surprising, because both $q_I$ are largely constrained by the same $Z_H$ observations. Secondly, retrieved $N_I$ from Murphy20 is much larger than that from ENCORE-ice. These results suggest that Murphy20 has attributed all radar signals to one species like our pristine ice. Due to the smaller size of pristine ice compared to aggregates, retrieved $N_I$ from Murphy20 must be much larger than ENCORE-ice to make up for the same $q_I$. This also explains why $N_I$ and $q_I$ in Murphy20 retrievals have similar profile shapes. Considering that the observed $\rho_{HV}$ is not close to 1, the attribution to single
species is likely inappropriate, leading to a large error in ice number concentration, even though $q_I$ may seem reasonable.





Finally, $D_{\mathrm{eff,mlt}}$, the effective mean diameter using the equivalent melted diameter as the size descriptor, from Murphy20 tends to be larger than that from ENCORE-ice. This is partly because Murphy et al. (2020) have used denser ice particle, i.e., the pre-factor and the exponent in their mass-size relationship are both slightly larger than our aggregates. Since $D_{\mathrm{eff,mlt}}$ depends on the assumed mass-size relationship, Fig. 8(l) is used for a qualitative comparison only.

Extending the evaluation from one ray to collocated data, a set of thresholds for matching space and time is needed. Since the enhanced area in Fig. 6 is about 15-km wide and 1-km deep, we use this scale as one of our criteria and consider in-situ observations and radar gates collocated if their distance is within 15 km in the horizontal and 1 km in the vertical. The time difference threshold for collocation is set to be within 3.5 mins, because a pair of back-to-back radar RHI scans were performed every 7 mins (see Sec. 2). These spatial and temporal thresholds lead to six clusters of radar scans for

intercomparison, which comprises 105 rays with a total of 1675 gates. For a given ray, if the root-mean-square-difference between the measured and the forward simulated radar observable is greater than 0.1 dB in $Z_{\mathrm{DR}}$, 0.1 ° km$^{-1}$ in $K_{\mathrm{DP}}$, or 0.01 in $\rho_{\mathrm{HV}}$, we consider that retrieval quality for the entire ray is poor and exclude all the retrievals. After this exclusion, 81 rays with 1237 radar gates remain for the following evaluation.

Figure 9 shows the time series of in-situ observations and collocated retrievals. Based on the measured temperatures, we

expect column ice crystals in the beginning and very end of the time series, because of associated temperature zones warmer than –10°C. The flight height was maintained at ~2 km from 6:30 to 6:40 UTC, suggesting that the temperature dip at 6:40 UTC is a data glitch, and that the temperature is likely to be about –5°C in reality. During 7:10–8:45 UTC, the temperatures are between –10°C and –20°C and likely favour the presence of both dendrite and plate. These expectations about prevalent ice habits are confirmed by in-situ cloud particle images (see Fig. 1 for examples), and generally consistent with our results.

In our retrievals, 40% of the collocated radar observables are best fit with plate as the pristine ice habit, 20% with dendrite, and 40% with columns that occur more frequently in the beginning and the end of the time series. Between plate and dendrite habits, our retrievals have shown a dominance of plate. Unfortunately, this is inconsistent with particle images that show a dominance of dendrite over plate. Therefore, we consider there remains a large uncertainty when distinguishing plate and dendrites, although the examinations of agreement between the measured and forward simulated radar observables

clearly indicate that the choice of plate was appropriate. Note that even with this habit uncertainty, the choice of plate and dendrite does not lead to significantly different retrievals in $N_{\mathrm{I}}$ and $q_{\mathrm{I}}$.

The collocated retrievals in Fig. 9c and 9d show that $N_{\mathrm{I}}$ retrieved from ENCORE-ice is approximately in the same order of magnitude as observations, and that the retrieved $q_{\mathrm{I}}$ values are close to the Nevzorov probe observations. $N_{\mathrm{I}}$ and $q_{\mathrm{I}}$ from ENCORE-ice generally perform better than those from Murphy20, but they are both overestimated as indicated by the box

plots in Fig. 10. The overestimations in the median of $N_{\mathrm{I}}$ and $q_{\mathrm{I}}$ are respectively 98% and 44% for ENCORE-ice, and respectively 445% and 187% for Murphy20. Note that Murphy20 retrievals in $N_{\mathrm{I}}$ and $q_{\mathrm{I}}$ are based empirical relationships derived from size bins between zero and infinity, while HVPS- and ENOCRE-ice-based estimates are derived using HVPS size bins from between 75 and 19275 μm. This difference in size ranges is not a concern for comparisons in $q_{\mathrm{I}}$, but it





contributes to part of the overestimation in $N_I$ in Murhpy20 retrievals. Using our retrieved PSD, we have found that the median $N_I$ derived from zero and infinity size bins is ~3% larger than that derived from the HVPS size range. This suggests that the difference in the size range for integration calculations is not the main cause for the 445% overestimation in Murphy20 $N_I$ retrievals. Additionally, similar to Fig. 8(l), Murphy20 $D_{eff,mlt}$ tends to be larger than our $D_{eff,mlt}$ from aggregates by 0.3 mm in the overall median, as shown in Fig. 9(f) and 10(d).

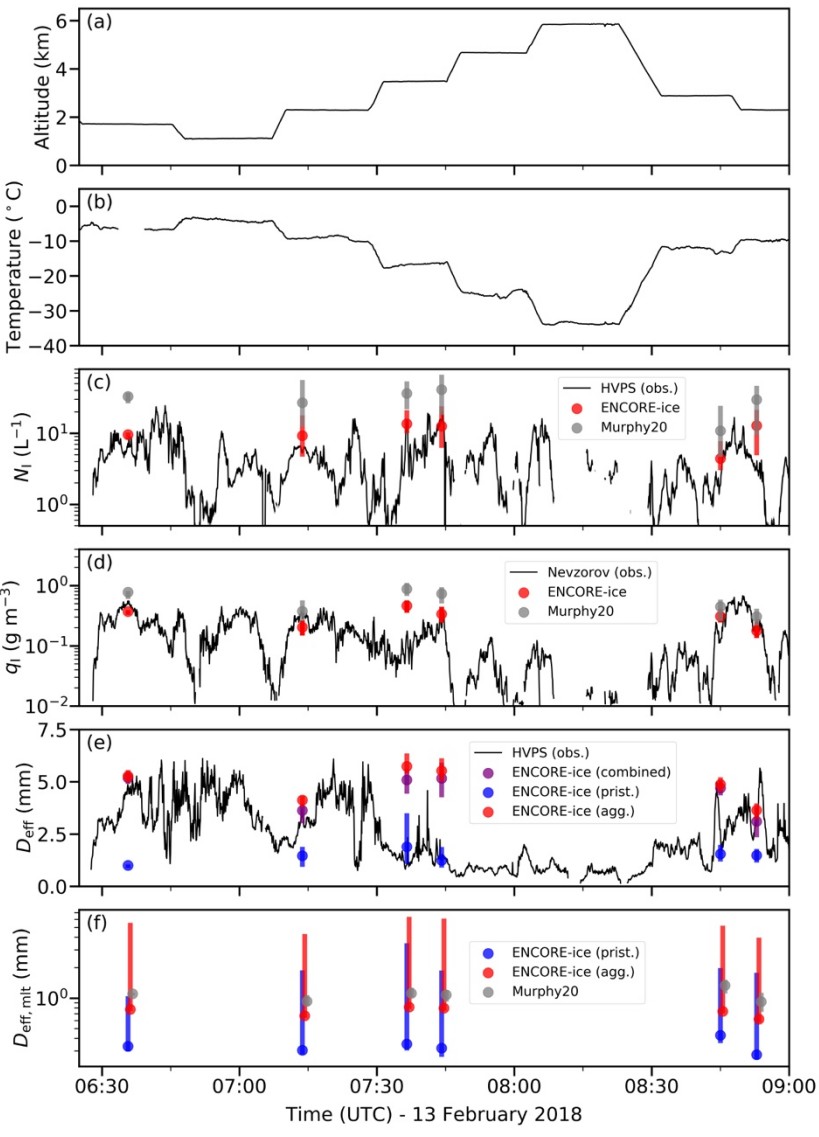

**Figure 9.** Time series of (a) flight altitude, (b) temperature, observed and retrieved (c) total ice number concentration, (d) ice water content, (e) effective mean diameter using the maximum particle dimension as the size descriptor, and (f) effective mean diameter using the equivalent melted diameter as the size descriptor. Retrieval from ENCORE-ice and Murphy20 empirical relationships are denoted by dots, explained by detailed legends. The dots represent the median of retrieval from all collocated gates, and the vertical bars denote the range between the 25th and 75th percentiles. Note that the counting uncertainty in total ice number concentration in (c) is plotted but too 480 small to see. All calculations are based on the size range of HVPS observations, except Murphy20 retrievals in (c) and (d).

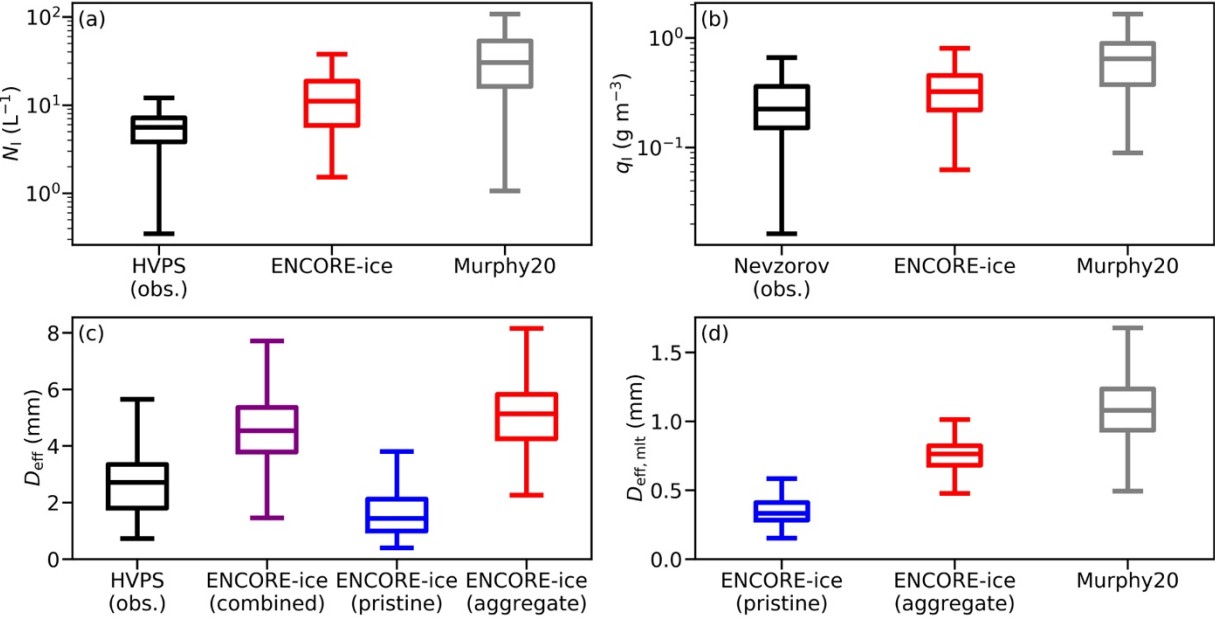

**Figure 10.** Box plots of in-situ observations and retrievals from ENCORE-ice and Murphy20 for (a) total ice number concentration, (b) ice water content, (c) effective mean diameter using the maximum particle dimension as the size descriptor, and (f) effective mean diameter using the equivalent melted diameter as the size descriptor. The bottom and top of each box represent the 25% and 75% quartiles, and the line inside the box represents the median. The whiskers mark represents the range of data points within 1.5 times the interquartile distance. The included sample sizes for in-situ data and radar gates are 347 and 1237, respectively.

Recall per equations (6) that the effective mean diameter is weighted by size and thus strongly influenced by large particles. When observations sample both pristine ice and aggregates, the combined mean particle size is expected to be close to the size of aggregates, not pristine ice. The results in Fig. 9(e) generally match this expectation, showing a reasonable agreement between the observed and the combined mean diameter, except three data clusters during the period between 7:30 and 8:45 UTC. For the cluster around at 8:45 UTC, $D_{eff}$ from HVPS has a large variation, ranging between 1.4 and ~5 mm with a median of 2.8 mm. This range is in the same order of $D_{eff}$ of pristine ice (1.5 mm) and of aggregates (~5 mm), though the median combined $D_{eff}$ of 4.7 mm is significantly larger than the observed median. This suggests that there may be a need to find a more robust way for size comparisons. For the clusters between 7:30 and 7:45 UTC, the observed effective mean diameter is closer to the retrieved pristine ice diameter. This unexpected behaviour might suggest a few scenarios. The first scenario is that the radar volume might include pristine ice only or aggregates only. However, as shown in Murphy20 retrievals, assuming single species leads to a large error in $N_P$. The observed $\rho_{HV}$ is also too low to support this scenario. The second scenario is that the separation between pristine ice and aggregates in our retrieval is inappropriate. To assess this possibility, we tested various combinations of $N_0$ and $D_0$, and found that the following two conditions must be met for the combined size to be close to the size of pristine ice. The first is that $N_P$ needs to be at least one order of the magnitude larger than $N_A$, and the second is that $D_{0,P}$ cannot be much smaller than $D_{0,A}$. To meet the first condition, let us assume that our retrieved $N_A$ is supposed to be 10 times smaller, because our retrieved $N_P$ from ENCORE-ice is already





overestimated compared to the in-situ observations and should not be even higher. Under that assumption, $D_{0,A}$ needs to
increase by a factor of 1.5 to maintain the same radar reflectivity observation. An increase in $D_{0,A}$, however, would make the
difference between $D_{0,P}$ and $D_{0,A}$ even larger, which violates the second required condition. The combination of reduced $N_A$
and increased $D_{0,A}$ by the factors used above would also reduce $q_I$ by a factor of 3. Then, to make up for the reduction of $q_I$,
one can increase $D_{0,P}$ to remediate the second required condition. This eventually leads to a scenario that two species are
alike, as the first scenario, which is not supported by the observations. The third scenario is that we have been focusing on
radar rays with reduced $\rho_{HV}$, and enhanced $Z_{DR}$ and $K_{DP}$, which may be only a portion of the samples that aircraft sampled.
Further studies using more datasets and retrievals would be needed to better explain why the observed effective mean
diameter is close to the retrieved size of pristine ice particle between 7:30 and 7:45 UTC. Overall, when including clusters
before 7:30 and after 8:45 UTC, the difference between the observed and the retrieved combined median $D_{eff}$ is about 0.55
mm.

## 5 Summary

We have introduced a new method for retrieving microphysical properties of concurrent pristine ice and snow aggregates
from X-band polarimetric radar observations. The radar observables used here include horizontal reflectivity, differential
reflectivity, co-polar correlation coefficient, and specific differential phase shift. The first observable provides constraints on
aggregate properties, while the last three observables provide constraints on the partitioning between aggregates and pristine
ice, as well as on the ice number concentration and size of pristine ice. The observations are combined with our prior
knowledge via an ensemble retrieval framework to find the best estimates of microphysical properties. Since properties of
pristine ice and snow aggregates vary significantly in nature, we apply a wide-spread prior and thus the retrieval is mainly
dominated by the observations.

Based on the evaluation using synthetic observations, we have found that the current observational uncertainty is
sufficient for quantifying properties of pristine ice and snow aggregates. The retrieval was able to reproduce vertical profiles
similar to the truth, and the root-mean-square-error with respect to the truth is within the retrieval uncertainty. The biases in
the combined total ice number concentration, ice water content, and effective mean diameter are all within 5%. This exercise
demonstrates that our retrieval method works well, if the prior and the forward models for simulating radar observables are
chosen appropriately and representative of reality. In general, the appropriateness and representativeness of the prior and
forward model can be confirmed by examining the agreement between the observations and the forward simulations.

We have also evaluated our retrieval against in-situ cloud probe observations taken from a recent field campaign in
Chilbolton, UK, which was coordinated to have collocated X-band radar scans and aircraft flights. We analysed a three-hour
long case that had 1237 collocated radar gates. Although the period was not particularly long, the aircraft sampled ice
particles in temperature zones from –5°C to –35°C, allowing us to assess the retrieval performance for cases that are
dominated by column, plate, or dendrite. The collocated in-situ data has a median number concentration of 5.6 L$^{-1}$, ice water



content of 0.2 g m$^{-3}$, and 2.7 mm effective mean diameter. Compared to in-situ medians, our retrieved total number concentration and ice water content are overestimated by 98% and 44%, respectively. This performance is generally better than that from empirical relationships, which has differences of 445% and 187% in total number concentration and ice water content, respectively with respect to the in-situ medians. For effective mean diameter, the in-situ observations agree with our

effective mean diameter combined from pristine ice and aggregate in three data clusters with a difference of 0.55 mm. However, in other clusters, the observed effective mean diameter can agree better with the retrieved size of pristine ice, which does not match the fact that the effective mean diameter should be dominated by larger aggregate particles. Further studies would be needed to explain why this occurs and to identify possible improvement in retrieved effective mean diameter.

This work is the first step toward quantifying microphysical properties of concurrent ice species, using a framework that considers our prior knowledge and the observational uncertainties. Since we have focused on radar signals with enhanced differential reflectivity and specific differential phase shift (i.e., cases with potentially high ice number concentration), the immediate application will be on studying secondary ice production. In particular, the Atmospheric Radiation Measurement (ARM) Program User Facility has operated X-band polarimetric radars at a fixed site at Barrow, Alaska, and in the Biogenic

Aerosols–Effects on Clouds and Climate field campaign in Finland back in 2014. These rich datasets will allow us to document how frequently high ice number concentrations occur, and how much larger these concentrations are compared to the number of primary ice particles. This can be further compared to model simulations to understand what controls the ice number productions.

**Appendix A**

Radar equations for a single sample volume containing multiple ice particle habits are given as:

$$Z_h = \frac{4\lambda^4}{\pi^4|K_w|^2}\sum_{i=1}^{J}\int_0^{\infty}\overline{\left\{A\left|S_{hh,\iota}^b\right|^2 + B\left|S_{vv,\iota}^b\right|^2 + 2CRe\left[S_{hh,\iota}^b S_{vv,\iota}^{b^*}\right]\right\}}n(D)dD; \tag{A1}$$

$$Z_v = \frac{4\lambda^4}{\pi^4|K_w|^2}\sum_{i=1}^{J}\int_0^{\infty}\overline{\left\{B\left|S_{hh,\iota}^b\right|^2 + A\left|S_{vv,\iota}^b\right|^2 + 2CRe\left[S_{hh,\iota}^b S_{vv,\iota}^{b^*}\right]\right\}}n(D)dD; \tag{A2}$$

$$Z_{hv} = \frac{4\lambda^4}{\pi^4|K_w|^2}\sum_{i=1}^{J}\int_0^{\infty}\overline{\left\{C\left[\left|S_{hh,\iota}^b\right|^2 + \left|S_{vv,\iota}^b\right|^2\right] + A\left[S_{hh,\iota}^b S_{vv,\iota}^{b^*}\right] + B\left[S_{vv,\iota}^b S_{hh,\iota}^{b^*}\right]\right\}}n(D)dD; \tag{A3}$$

$$Z_{dr} = \frac{Z_h}{Z_v}; \tag{A4}$$

$$\rho_{hv} = \frac{|Z_{hv}|}{[(Z_h)(Z_v)]^{1/2}}; \text{ and} \tag{A5}$$

$$K_{DP} = \frac{0.18\lambda}{\pi}\sum_{i=1}^{J}\int_0^{\infty}\left\{C_k Re\left[S_{hh,\iota}^f - S_{vv,\iota}^f\right]\right\}n(D)dD, \tag{A6}$$





where $Z_h$, $Z_v$, and $Z_{hv}$ are in units of mm$^6$ m$^{-3}$; $D$ is the maximum particle dimension; $\lambda$ is the radar wavelength; $K_w$ is the dielectric factor of water and $|K_w|^2 = 0.93$; and the amplitude scattering matrix elements ($S$) are in units of mm. The vertical bars represent the magnitude of the terms within, while $Re$ represents the real part of the complex number and the asterisk

indicates its complex conjugate. The index $i$ represents the species existing in the radar volume, and the index $J$ represents the number of species. Note that the amplitude scattering matrix elements in the database are tabulated for various elevation angles, azimuth angles, and habit realizations. To apply these amplitude elements to the equations (A1)–(A6), $S_{hh}^{f,b}$ and $S_{vv}^{f,b}$ are first linearly interpolated with respect to elevation angles for radar rays. The interpolated $S_{hh}^{f,b}$ and $S_{vv}^{f,b}$ are then used to calculate the terms in the parentheses for all azimuth angles and habit variations. Because the azimuthal orientation of

hydrometeors relative to the radar is random and unknown, and because the exact morphological characteristics of these particles at any given time in nature are also unknown, the terms in the parentheses are averaged over azimuth angles and habit realizations, which are represented by the horizontal bar over the parentheses.

Coefficients $A$, $B$, $C$, and $C_k$ are included to account for the effects of canting on the polarimetric radar moments. Following Jung et al. (2010) and Ryzhkov et al. (2011), the canting angle distributions are assumed to be Gaussian, and their

effects can be parameterized using the mean and standard deviation of the distribution. Supposing that all oblate species fall with their major axes preferentially oriented in the horizontal plane, the mean canting angle can be set to zero (Ryzhkov et al. 2011). The width of the canting angle distribution is set to 10° for pristine ice crystals and 60° for snow aggregates, similar to Ryzhkov et al. (2011) and Matsui et al. (2019). All detailed equations and coefficients can be found in Jung et al. (2010).

**Appendix B**

Ryzhkov et al. (2018) used a power-law dependence to describe particle density, given as:

$$\rho = \alpha D_e^{-1}, \tag{B1}$$

where the density $\rho$ is in g cm$^{-3}$, coefficient $\alpha$ is in g cm$^{-2}$, and $D_e$ is the equivalent volume diameter. They also assumed an exponential particle size distribution, i.e.,

$$N(D_e) = N_{0,s}e^{-\Lambda D_e}, \tag{B2}$$

with an intercept $N_{0,s}$ and the exponent $\Lambda$. From their equations (3) and (4) in Ryzhkov et al. (2018), we can calculate these two parameters by:

$$\Lambda = \frac{4}{D_{emp}}, \tag{B3}$$

$$\alpha = 0.00309 \frac{Z}{q_{I,emp} \cdot D_{emp}^2}, \text{ and} \tag{B4}$$



$$N_{0,s} = \frac{q_{I,emp}}{0.0003811 \cdot \alpha^{-0.2} Z^{0.6}}, \tag{B5}$$

where the exponent $\Lambda$ in mm$^{-1}$, $N_{0,s}$ in m$^{-3}$ mm$^{-1}$, $\alpha$ is in g cm$^{-2}$, $Z$ is the radar reflectivity in mm$^6$ m$^{-3}$, $q_{I,emp}$ is the retrieved ice water content from equation (19) in g m$^{-3}$, and $D_{emp}$ is the retrieved diameter from equation (17) in mm.

Once $N_{0,s}$ and $\Lambda$ are known in equation (B2), we further convert the size descriptor $D_e$ to the equivalent melted dimeter (denoted as $D_{mlt}$) by

$$D_{mlt} = \left(\frac{\alpha}{\rho_w} D_e^2\right)^{\frac{1}{3}}, \tag{B6}$$

and then calculate the effective mean diameter $D_{eff,mlt}$ using equation (7).

*Data availability.* FAAM aircraft observations from PICASSO are available at the Centre for Environmental Data Analysis archive (https://www.ceda.ac.uk/). NxPol radar observations from PICASSO are publicly available via https://catalogue.ceda.ac.uk/uuid/ffc9ed384aea471dab35901cf62f70be. The ice crystal scattering database used to compute radar moments is available at https://www.arm.gov/data/data-sources/icepart-mod-120. The retrieval will be available freely in the ARM Archive as a PI product.

*Author contributions.* All authors contribute to the work presented here and manuscript editing. NK analysed the radar and aircraft observations, coded the algorithm, performed the retrievals, and provided the initial draft. JC supervised the project, conceptualized the idea, developed the methodology, analysed the retrievals, and revised the manuscript. SB, SJ, and VC quality controlled the radar observations and contextualized observed polarimetric features. YL provided guidance in constructing the radar forward model. PJvL introduced the most recent development in the Iterative Stochastic Ensemble Kalman approach and assisted in improving the retrieval algorithm. CW provided and contextualized the PICASSO observations. YB assisted in data analysis and retrieval preparation. SO provided and quality controlled the in-situ cloud probe data.

*Competing interests.* The authors declare that they have no conflict of interest.

*Acknowledgements.* This research was supported by the Office of Science (BER), DOE under grants DE-SC0018930. PJvL was sponsored by the European Research Council via the CUNDA project under number 694509. CW and SO were supported by the Natural Environment Research Council (NERC) under grant number NE/P012426/1. Airborne data were obtained using the BAe-146-301 Atmospheric Research Aircraft (ARA) flown by Directflight Ltd and managed by the Facility for Airborne Atmospheric Measurements (FAAM), which is a joint entity of NERC and the Met Office. This work



would not be possible without the efforts of the FAAM group and the PICASSO research team led by Jonathan Crosier. We also thank Ryan Neely III and Lindsay Bennett at NCAS/University of Leeds for collection of the NXPOL radar data.

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
