# Peer review of "Retrieving microphysical properties of concurrent pristine ice and snow using polarimetric radar observations"

_Atmospheric Measurement Techniques, 2021_

## Author Response (AR2)

**Response to Reviewer #1**

1. Summary: This paper presents a framework for estimating the bulk properties of pristine and aggregated ice particles from polarimetric radar measurements. The framework retrieval methods for ice number concentration and ice water content, however, retrievals of the effective mean diameter are less accurate consists of an iterative ensemble algorithm to estimate the distribution of particle size distribution (PSD) parameters by informing a prior distribution with radar measurements. The radar measurements are simulated from the PSD parameters using scattering database results for several types of ice particles. The retrieval method is first tested in a known truth experiment, with retrieved parameters that agree with the true parameters. Observed radar data are then used to demonstrate the retrieval, and in situ aircraft measurements are used for validation on a number of radar gates. The results improve upon previous empirical.

   I find this paper to be well written overall and I think it should advance the use of radar measurements to understand ice precipitation. I especially appreciate that the method presented in the manuscript provides uncertainty estimates for the retrieved quantities and the authors do a nice job incorporating the retrieval uncertainty into their analysis. I do have a few general comments and several specific comments that should be addressed, and therefore I find this manuscript acceptable for publication subject to minor revisions.

   We thank the reviewer for thoughtful comments.

**General comments**

2. I have some concerns with the collocation of the in situ aircraft measurements and the radar data. It is perhaps reasonable to have a relatively large acceptable horizontal distance between the aircraft and the radar measurements, subject to the condition that the precipitation is horizontally homogeneous. However, discrepancies in the vertical positions of these measurements are less acceptable. Within a layer favorable for planar crystal growth and subsequent aggregation, there will typically be large vertical gradients in the radar measurements; these gradients reflect the rapid growth via vapor deposition and large changes in the bulk particle properties (e.g., mean diameter) via aggregation. Given these large gradients, measurements offset vertically from each other can represent substantially different ice particle populations. The authors need to address this collocation threshold by reducing the acceptable vertical difference between the in situ and radar measurements. Without knowing the relative locations of these observation sources, it is difficult to draw conclusions about why the retrieved effective diameters do not agree with the measurements.

   We understand the reviewer's concern. Since this comment is related to the Reviewer's Specific Comment #30 about how well the collocation is (see Page 14), we will provide combined responses to explain our findings and decision. For convenience, we have taken part of Fig. 9 from the manuscript and included here as Fig. R0.

   Recall that our retrieved ice number concentration and ice water content perform well compared to in-situ data, but the performance in effective diameter ($D_{eff}$) is puzzling. As shown in Fig. R0, for Clusters 1, 2 and 6, in-situ $D_{eff}$ is closer to the retrieved $D_{eff}$ of aggregates, which matches our expectation because aggregates will dominate the combined $D_{eff}$. However, for Clusters 3–5, in-situ $D_{eff}$ is closer to the retrieved $D_{eff}$ of pristine ice, and we had difficulty to understand why that's the case. We thank the reviewer for this question, because it may have led to the most plausible explanations for this puzzling behavior in $D_{eff}$.

To assess the impact of the collocation threshold in the vertical, we decrease its value from 1 km to 0.5 km (see Fig. R1). Comparing Figs. R0 with R1, we see that the stricter threshold removes the Cluster 1, slightly improves Cluster 6 in ice number concentration retrieval, but the overall picture and error statistics remain similar to our previous results.

We have decided to keep our original figures and statistics because of three reasons. Firstly, our results are not sensitive to the reduction of the collocation threshold in the vertical. Secondly, Cluster 1 is important to be included to represent retrievals in conditions dominated by columns. Thirdly, the current collocation leads to some interesting results that link sampling to cloud processes, as shown below.

To understand how well the collocation is, we have produced 2D histograms of occurrences of the vertical and horizontal distance in the collocated in-situ and radar dataset, as shown in Fig. R2. The distance was calculated with respect to radar gate, i.e., the positive vertical distance represents that the flight altitude is higher than the radar gate of interest.

Checking the collocation in Fig. R2, we found that in-situ samples were taken largely at radar scan heights or below in Clusters 1, 2 and 6. It is likely that both in-situ and radar have sampled the same regime with notable aggregations, and that's why the observed $D_{\text{eff}}$ is close to the retrieved $D_{\text{eff}}$ of aggregates. In contrast, in-situ samples were taken at higher altitudes over the radar scans in Clusters 3–5. As the reviewer suggested, we believe that in Clusters 3–5, aircraft may have sampled a pristine ice growth zone aloft, but the radar gates below sampled the subsequent aggregations, which explains why the observed $D_{\text{eff}}$ is closer to the retrieved $D_{\text{eff}}$ of pristine ice, rather than aggregates.

In the revised manuscript, we now have included Fig. R2 as Fig. 11 and added the following text:

Lines 528–536:
The third scenario is that the discrepancy in $D_{\text{eff}}$ is due to a sampling issue. Figure 11 shows two-dimensional histograms of occurrences of the vertical and horizontal distance in the collocated in-situ and radar dataset. The distance was calculated with respect to radar gate, i.e., the positive vertical distance represents that the flight altitude is higher than the radar gate of interest. Interestingly, for Clusters 1, 2 and 6, in-situ samples were taken largely at radar scan heights or below. It is likely that both in-situ and radar have sampled the same regime with notable aggregations, which explains why the observed $D_{\text{eff}}$ is close to the retrieved $D_{\text{eff}}$ of aggregates. In contrast, in-situ samples were taken at higher altitudes over the radar scans for Clusters 3–5. In these cases, aircraft may have sampled a pristine ice growth zone aloft, but the radar gates below sampled the subsequent aggregations, which explains why the observed $D_{\text{eff}}$ is closer to the retrieved $D_{\text{eff}}$ of pristine ice, rather than aggregates. Further studies using more datasets and retrievals would be needed to assess the third scenario.

Lines 569–574 in the Summary Section:
… In other clusters, the observed effective mean diameters agree better with the retrieved size of pristine ice, likely because the aircraft sampled pristine ice growth zones aloft instead of aggregation zones that radar sampled. Since planar crystal growth and subsequent aggregation can lead to zones with distinct ice bulk properties, taking frequent aircraft measurements at multiple vertical layers around the radar location would be particularly helpful to improve collocations and allow us to analyse individual rays in more detail.

[Figure]

***Figure R0:*** *Excerpted from Fig. 9 in the manuscript. Time series of (c) total ice number concentration, (d) ice water content, (e) effective mean diameter using the maximum particle dimension as the size descriptor, and (f) effective mean diameter using the equivalent melted diameter as the size descriptor. Retrieval from ENCORE-ice and Murphy20 empirical relationships are denoted by dots, explained by detailed legends. The dots represent the median of retrieval from all collocated gates, and the vertical bars denote the range between the 25th and 75th percentiles. For convenience, we index six retrieval clusters from 1 to 6 as shown in (d). Results are based on a collocation threshold of 1.0 km in the vertical.*

[Figure]

**Figure R1:** *Same as Fig. R0 but using a collocation threshold of 0.5 km in the vertical.*

[Figure]

**Figure R2:** *2D histograms of occurrences of distances in the vertical and horizontal between in-situ measurements and radar gates for Cluster 1–6 in (a)–(f), respectively. Note that occurrences are counted for all pairs of in-situ data point and radar gate. In calculations of retrieval errors, selected in-situ data points and radar gates are only used once with equal weights.*

3. Additionally, some of the details in the ensemble algorithm are unclear and I think require a bit more explanation to understand the algorithm and contextualize its results. For example, is there an assumed form for the prior and posterior (e.g., Gaussian), and are the state vector elements assumed to be independent in the prior? Do these assumptions about the prior probability distribution impacts the retrieval results?

Firstly, the prior is assumed Gaussian, but the posterior is not, although we do concentrate on the mean and standard deviation of the posterior.

Secondly, as described as the "second consideration" in Section 2.3, there is no prior correlation between variables, but there is correlation in the vertical for each variable via the slopes and an auto-regressive order 1 process. Therefore, no, the state vector elements are not completely independent in the prior.

Thirdly, the prior is chosen wide, approximately 1–2 orders of magnitude in the state variables, such that the influence of the prior is minimal.

Although these details are provided in Section 2.3, we agree that some information can be added in Section 2.2.2 to help readers to understand the algorithm. We now include the following text:

> Lines 157–161:
> As detailed later in Section 2.3, the prior is assumed Gaussian, and there is no prior correlation between variables $N_{0,P}$, $D_{0,P}$, $N_{0,A}$, and $D_{0,A}$. But there is correlation in the vertical (i.e., between gates) for each variable in our setup. We have also used a prior with large uncertainty, approximately 1–2 orders of magnitude in the state variables, such that the influence of the prior is minimal. In contrast to the prior, no Gaussian assumption is made in the posterior ensemble members, although the retrieval statistics are largely focused on their means and standard deviations.

4. Finally, there should be more discussion regarding the applicability of the retrieval method in various ice growth regimes. The authors do mention using the framework to study secondary ice nucleation; however, it is unclear whether this application would require also considering rimed particles. In general, rimed particles, aggregates, and pristine ice crystals of different habits can all exist in the same radar sample volume. This manuscript would benefit from a more thorough discussion about the current capabilities to handle these more complex situations and/or future plans to implement them.

We have included the following text in the Summary Section, and revised the last paragraph to provide better links to ice growth regimes:

> Lines 575–582:
>
> Currently, our method is designed to work for conditions with a mixture of pristine ice and aggregates. In the presence of rimed particles, the state vector should be expanded to include additional variables that can accommodate and inform the degree of rimming, e.g., the riming factor described in Masson et al. (2018), or to include appropriate rimed species explicitly. When triple-frequency measurements are available and can be used to distinguish particle types effectively (e.g., Kneifel et al., 2015; Barrett et al., 2019), such information on particle types can also be incorporated into our method to provide retrievals for off-zenith radar scans that are more challenging for triplefrequency techniques. It is also possible to expand the observation vector with other radar observables at multiple wavelengths, providing further constraints on retrieval if added information exists.

This work is the first step toward quantifying microphysical properties of concurrent ice species, using a framework that considers our prior knowledge and the observational uncertainties. Since we have focused on radar signals with reduced co-polar correlation coefficient and enhanced differential reflectivity and specific differential phase shift (i.e., cases with potentially high ice number concentration), the immediate application will be on studying dendritic growth zones commonly found in thick stratiform clouds. In particular, the Atmospheric Radiation Measurement (ARM) Program User Facility has operated X-band polarimetric radars at a fixed site at Barrow, Alaska, and in the Biogenic Aerosols–Effects on Clouds and Climate field campaign in Finland back in 2014. These rich datasets will allow us to study formation of new crystals either via primary nucleation or a secondary ice process, their growth into planar crystals and dendrites, and the subsequent aggregations. The retrieved ice properties can be further compared to model simulations to understand what controls the ice number productions.

**Specific comments**

5. Line 45-48: This sentence needs to be revised to more clearly motivate why understanding the relative proportions of aggregates and pristine ice crystals is important in understanding cloud and precipitation processes.

   Apologies that those references were not cited properly. We have revised the sentence as:

   Lines 45–46:
   The ability of the partitioning is of particular importance for studying the aggregation process, because it provides information on size and number concentration of pristine ice and aggregates.

6. Lines 76-77: It is important to note here that aggregates have an infinite variety of complex shapes. Characterizing them as having "spheroidal morphologies" is incorrect, and stating that ZDR is low because particles are spheroids is also incorrect because highly oriented, high-density spheroids can have high ZDR. Please rework this explanation for the assumed ZDR of aggregates.

   We have revised the text as:

   Lines 75–76:
   Snow aggregates yield low $Z_{DR}$ (about 0–0.6 dB; see Hogan et al., 2012) as a result of their sparse and irregular morphology, with the component crystals oriented at a wide range of angles.

   Lines 739–740:
   Hogan, R. J., Tian, L., Brown, P. R. A., Westbrook, C., Heymsfield, A. J. and Eastment, J. D.: Radar scattering from ice aggregates using the horizontally aligned oblate spheroid approximation. J. Appl. Meteorol. Clim., 51, 655–671, doi: https://doi.org/10.1175/JAMC-D-11-074.1, 2012.

7. Line 86: What do you mean by the statement that Rhohv and KDP are "more advanced" measurements? Please be more specific.

We apologize for the confusing wording. We meant that $\rho_{HV}$ and $K_{DP}$ observations have not been used extensively in quantitative retrievals. Since the other reviewer also has the same question, and the information provided here was not critical, we have removed this sentence.

8. Lines 106-107: The size distribution must include the differential size (i.e., dD) in order to represent the number concentration of particles within the differential size range about D. Please modify.

Agreed. We have revised the sentence to:

Lines 103–104:
… and $n(D)dD$ is the number of particles in the range of the maximum particle dimensions $(D, D + dD)$.

9. Lines 114-122: It's a bit unclear whether there are separate PSDs for the aggregates and pristine ice crystals since the equations here are written in terms of a single PSD (n(D)), but later in the manuscript N0 and D0 are retrieved for both aggregates and pristine ice crystals. Please clarify in this section.

Agreed. We have re-written equation (3) on Line 113 as:

$$N_I = \int_0^\infty n(D)dD = \int_0^\infty [n_P(D) + n_A(D)]dD = \int_0^\infty N_{0,P}f_\mu(D; D_{0,P})dD + \int_0^\infty N_{0,A}f_\mu(D; D_{0,A})dD = N_P + N_A,$$

and have added the following text:

Lines 115–116:
where $n(D)$ is the combined PSD from $n_P(D) + n_A(D)$, and the subscripts P and A denote contributions from pristine ice and snow aggregates, respectively.

10. Lines 145-150: Clarify whether the state vector is updated independently at each range gate. In other words, do radar observations at one range gate influence the estimation of the state vector at another gate?

As described on Lines 263–273 in Section 2.3, radar observations at one range gate will influence the estimation of the state vector at another gate, if these two gates are within the pre-defined radius. We have added the following text right after we introduce the observation vector:

Lines 151–152:
Radar observations at one range gate will influence the estimation of the state vector at another gate, if these two gates are within the pre-defined radius, which will be explained in more detail in Section 2.3.

11. Line 152: Are there values of KDP less than or equal to zero? If so, how are they handled when logarithms of these values are taken to create the observation vector?

   As mentioned on Line 285 (Now Line 297 in the revised manuscript), we excluded gates with $K_{DP}$ below 0.1° km$^{-1}$ to ensure a sufficient number concentration of pristine ice. This exclusion also ensures that it is OK to take logarithms of $K_{DP}$.

12. Lines 180-181: The aggregates in the Lu et al. (2016) database have somewhat limited shapes since the monomers they are composed of are either single-width (in terms of a single GMM sphere) columns or stellar crystals with single-width branches. The authors should add some note here that natural aggregates may have substantially different properties for a given size compared to these simplified particles.

   Agreed. We have added the following text:

   > Lines 182–183:
   > Note that for a given size, natural aggregates may have substantially different properties compared to the realizations available in the database.

13. Line 196-197: Add a reference to support this statement.

   The reference we had in mind for that statement was:

   Dunnavan, E. L., Jiang, Z., Harrington, J. Y., Verlinde, J., Fitch, K., and Garrett, T. J.: The shape and density evolution of snow aggregates, 76, 3919–3940, 2019.

   Since we rewrote the paragraph to address reviewer's Comment #16, this sentence is no longer needed and have been deleted.

14. Lines 200-201: Clarify whether this mass-size relation corresponds to a pristine particle or an aggregate of pristine particles

   Thank you for pointing this out. It is for aggregates composed of ordinary dendritic crystal. We have revised the text as the following:

   > Lines 208–211:
   > In the mass-size relationship for LDt-P1d we used $a$= 0.000482 and $b$= 1.97 in units of cgs as in Table 2, based on aggregates composed of ordinary dendritic crystal (Kajikawa, 1989; Botta et al., 2011), whereas for HD-P1d we used $a$= 0.00145 and $b$ = 1.80 in units of cgs, based on aggregates of thin plate (Mitchell and Heymsfield, 2005; Botta et al., 2011).

15. Line 209: I do not see clear evidence of individual dendrites in Fig. 1c. It is more plausible that aggregates of dendrites present. Please update the figure caption to more precisely describe the particle types that can be inferred from that image.

   Agreed. We have changed the figure caption as:

> Lines 216–218:
> **Figure 1.** Examples of particle images from the Stratton Park Engineering Company Two-Dimension Stereo (2DS) probe, showing the presence of (a) column, (b) plate and (c) aggregates of dendrites on 13 February 2018. Each image frame is 1.28 mm high, taken from one of the probe channels only since the other channel was not working properly on this day.

16. Line 214: How are the axis ratios defined? Are they determined by fitting an ellipsoid to the aggregate shape? Please elaborate.

    Each aggregate in the database (Lu et al., 2016; see their Section 2.1) was generated by first specifying a reference spheroid with a given horizontal maximum dimension and an aspect ratio of 0.6, defined as the ratio of the lengths of the polar axes to the equatorial axes. Then, small monomers were added to the reference spheroid one at a time; any parts of the monomer that were outside the reference spheroid were removed. This procedure was repeated until the mass of the aggregate reached the desired total mass. As a result, the aspect ratio of the aggregate generated in the database was not necessarily the same as the reference spheroid (0.6).

    We now have included this information in the revised manuscript and reorganized some text, as shown below. We have also updated Fig. 2 in which the average ratio is now obtained by computing the ratio for each aggregate first and then taking an average. The original figure was done by computing the average horizontal maximum dimension and the average vertical maximum dimension first, and then taking the ratio. The revised method is more appropriate, although results from two methods are very similar.

> Lines 197–208:
>
> Similarly, the scattering database provides five types of aggregates; two of them were constructed using ice columns (LD-N1e and HD-N1e), three of them using stellar ice crystals (LD-P1d, LDt-P1d and HD-P1d). Each aggregate in the database (Lu et al., 2016) was generated by first specifying a reference spheroid with a given horizontal maximum dimension and an aspect ratio of 0.6, defined as the ratio of the lengths of the polar axes to the equatorial axes. Then, small monomers were added to the reference spheroid one at a time; any parts of the monomer that were outside the reference spheroid were removed. This procedure was repeated until the mass of the aggregate reached the desired total mass. As a result, the aspect ratio of the aggregate generated in the database was not necessarily the same as the reference spheroid (0.6).
>
> Figure 2 shows the average aspect ratios for aggregate types available in the database, which were calculated by averaging ratios of the maximum vertical dimension to the maximum horizontal dimension for all realizations within one size bin. Compared to Garrett et al. (2015) and Jiang et al. (2017) that reported an aspect ratio range between 0.3 to 0.6 from observations of falling aggregates at the surface, we found that LDt-P1d and HD-P1d exhibit a similar aspect ratio range.

17. Line 233: There is seemingly a contradiction here between stating that the variables other than N0p "lack a clear dependence on height" and line 242 where the slopes of D0P and D0A are assumed to be negative. Please clarify.

Sorry for the confusion. As explained on Line 244, since we use the logarithm of $D_0$ and $N_0$, we had to apply a slightly negative slope for for $D_{0,P}$, $N_{0,A}$, and $D_{0,A}$ to cover an appropriate range of state variables for radar gates at higher altitudes. Without this slightly negative slope, the prior at radar gates at higher altitudes can be unrealistic. Note that even with a slightly negative slope, the prior still contains many randomly generated zero and positive slopes. To clarity, we have reorganized this part as:

Lines 238–244:

… The slope was randomly selected from a normal distribution described in Table 4. Because the prevalence of active ice nuclei is a function of temperature and thus a function of height as well (DeMott et al., 2010), $N_{0,P}$ likely increases with height and thus the slopes in the prior are assumed to have a positive mean. In contrast, the dependence of $D_{0,P}$, $N_{0,A}$, and $D_{0,A}$ on height is less clear (e.g., Field et al., 2005). For practical reasons, the slopes applied for $D_{0,P}$, $N_{0,A}$, and $D_{0,A}$ are assumed to have a slightly negative mean. The slightly negative slope avoids unrealistic priors for radar gates at higher altitudes since we used the logarithm form in the state vector.

18. Lines 236-237: Please add some explanation for why you added noise in this way to the prior.

Without this noise term each ensemble member would be a straight line in the vertical for each variable with a different slope. Since the fundamental idea behind ensemble retrievals is that the true atmospheric profile is drawn from the same distribution as the prior ensemble members, and we know the true atmospheric profile is not a straight line, we add random noise with non-zero vertical correlation to each ensemble member profile to make each of them more realistic.

We have added this text in the revised manuscript on Lines 245–249.

19. Lines 243-244: Please clarify where the two order of magnitude range in the state variable prior comes from (i.e., is that comparing ensemble members over the entire vertical profile or at a specific range gate). It doesn't seem to correspond to the standard deviation values listed in the table, especially for the N0A and N0A

The two orders of magnitude came from comparing all ensemble members over the entire vertical profile. We have added this information in the revised manuscript:

Lines 260–261:
In general, the range in our prior is large, approximately 1–2 orders of magnitude across all ensemble members over the entire vertical profiles.

Note that the first row in Table 4 is given in the physical space, and the 2nd–4th rows are given in the log10 space. To demonstrate some back-of-envelope calculations, let us take $N_{0,P}$ as an example. $\log_{10}(50) = 1.7$. Let us use 1.7−2*sigma as the starting point at the lowest gate, which is equal to 1.7−2*0.15 = 1.4, i.e., $N_{0,P}$=25 L$^{-1}$ mm$^{-1}$. Assuming the associated slope=1−2*sigma = 1−2*0.2=0.6 (km$^{-1}$), then, at 1-km height, $\log_{10}(N_{0,P})$ = 1.4+0.6=2.0, i.e., $N_{0,P}$=100 L$^{-1}$ mm$^{-1}$. Now we repeat the same exercise for another point at the lowest gate, say 1.7+2*sigma = 1.7+2*0.15=2.0, i.e., $N_{0,P}$=100 L$^{-1}$ mm$^{-1}$. Assuming the associated slope = 1+2*sigma = 1+2*0.2=1.4 (km$^{-1}$), then at 1-km height, $\log_{10}(N_{0,P})$ = 2+1.4=3.4, i.e., $N_{0,P}$=2511 L$^{-1}$ mm$^{-1}$. To summarize, for the lowest gate, $N_{0,P}$ can vary from 25 L$^{-1}$ mm$^{-1}$ to 100 L$^{-1}$ mm$^{-1}$. At 1-km height, $N_{0,P}$ can vary from 100 L$^{-1}$ mm$^{-1}$ to 2511 L$^{-1}$ mm$^{-}$

$^1$. (spanning about 1.4 orders of magnitude). If we consider the range from 25 L$^{-1}$ mm$^{-1}$ to 2511 L$^{-1}$ mm$^{-1}$., then it is about a 2-order of magnitude difference.

20. Lines 285-287: The second sentence of this bullet point is incomplete. Please revise.

Thank you.  We have revised it as:

Lines 297–300:

… Note that negative $K_{DP}$ values indicate the presence of conical graupel (Aydin and Seliga 1984) or the vertical reorientation of pristine ice crystals in the presence of thunderstorm electric fields (Hubbert et al. 2014). Since our state vector only includes aggregates and horizontally orientated pristine ice, we exclude such gates as well.

21. Lines 312-315: Clarify whether the HVPS particle size distribution measurements are defined in terms of maximum diameter or volume-equivalent diameter (i.e., are they consistent with the PSDs used in the retrieval).

Yes, HVPS measurements are defined in terms of maximum diameter and are consistent with the PSDs used in the retrieval.  We have revised the text and included the information on the size descriptor of HVPS:

Lines 311–313:
The HVPS is an optical array particle imaging probe, which collects images of ice crystals with a pixel resolution of 150 µm. Size distributions of particles between 75 and 19275 µm were derived from their images and reported here using the maximum particle dimension as the size descriptor.

22. Line 316: Does "simultaneous evaluations" mean comparisons for the Deff, IWC, and total ice number concentration? Please clarify.

Yes, that is what we meant.  We have revised the text:

Lines 328–329:
The evaluations in the total ice number concentration, ice water content, and effective mean diameter all together allow us to indirectly examine whether the partitioning between pristine ice and aggregates is appropriate.

23. Line 321: There is no Ryzhkov and ZrniÄ (2019) in the references; I think the authors are referring to the book "Radar polarimetry for weather observations" by Ryzhkov and ZrniÄ (2019)?. Please check that all the in-text citations match the reference list.

Apologies for missing this reference in our list.  We now have included it in the References section:

Lines 824–825:
Ryzhkov, A. V., and Zrnic, D. S.: Polarimetric microphysical retrievals, in Radar Polarimetry for Weather Observations, Springer, 435–464, doi:10.1007/978-3-030-05093-1_11, 2019.

24. Lines 366-367: These compensating effects are suggested because the estimated N0A is higher than the true N0A and the estimated D0A is lower than the true D0A . Can you show a scatter plot of these retrieved parameters from the ensemble? An inverse relation between these parameters would more clearly demonstrate the compensation effect to satisfy reflectivity.

We have produced a scatter plot of $D_{0,A}$ vs $N_{0,A}$ at the lowest gate for 500 ensemble members. As expected, an inverse relation is evident and indicates the compensating effects between these two variables.

[Figure]

**Figure R3:** *A scatter plot of $D_{0,A}$ vs $N_{0,A}$ at the lowest gate for 500 ensemble members.*

25. Line 399: Give the range of temperatures within this region rather than a single temperature.

We have changed the single temperature of –15°C to a range between –12°C and –18°C. The following has been included:

Lines 412–413:
Based on the temperatures measured by the aircraft (Fig. 7), this area is in a temperature zone approximately between –12°C and –18°C, and…

26. Lines 403-404: Isn't the habit predetermined before running the retrieval? The latter half of this sentence makes it seem like the retrieval is providing habit information independently. Please rephrase.

No, the habit is not predetermined.  As mentioned on Lines 195–197, we ran our retrieval algorithm for all three habits, and then selected the most appropriate one based on the agreement in the measured and forward simulated radar observables.  To make it clearer, we have made some small changes in this sentence:

*Lines 195–197:*
Currently, we do not predetermine the ice habit. Instead, we ran our retrieval algorithm for all three habits independently, and then selected the most appropriate one based on the agreement in the measured and forward simulated radar observables.

27. Line 442: These spatial thresholds seem too large in cases where the precipitation is less homogeneous than what is observed in the RHI for the previous case (Fig. 6). I think the vertical distance threshold is especially problematic since the radar variables and associated ice particle

properties can change substantially over 1 km (e.g., the observed and retrieved microphysical variables shown in Fig. 8).

See responses to the reviewer's General Comment #2 on Page 1–3.

28. Lines 445-448: Are there general reasons why ~20% retrievals for these rays failed to reproduce the observed radar variables? There should be some discussion of this point to illustrate conditions where the retrieval assumptions are not satisfied.

Typically, unsuccessful retrievals fail to pass the $Z_{DR}$ criterium of 0.1dB. In most cases, we believe the failure is because we have not incorporated an appropriate prior. To make the retrieval method work for those 20% unsuccessful retrievals, we may need to assume a different shape of profiles for the state variables. Unfortunately, we do not have good knowledge of those shapes. It would be great if future aircraft flights can help gather that information, by taking frequent multiple-layer flights around the site. We now have included this in the revised manuscript:

Lines 461–464:
Most unsuccessful retrievals are likely due to an inappropriate prior. To make the retrieval method work for those unsuccessful cases, we may need to assume priors with different shapes of vertical profiles. Unfortunately, we do not have good knowledge of those shapes and will need to rely on future campaigns to help gather this information by taking frequent multiple-layer flights around the radar site.

29. Lines 459-461: This section requires a bit more explanation. Does assuming plates are the pristine category in the retrieval provide a better correspondence between the measured and forward-simulated radar variables for all of the in situ measurements or just the ones where dendrites are observed? Also, how are the dominant pristine habits determined from the imagery?

Firstly, the dominant pristine habits referred in the manuscript were determined by visually checking the in-situ cloud particle images. For clarity, "visually checking" has been added in the revised manuscript.

Secondly, when the cloud particle images were dominated by columns, indeed, we have also found that retrievals with columns as the pristine ice habit provide a better agreement between the measured and forward-simulated radar variables.

Thirdly, we saw dendrites much more frequently than plates in cloud particle images between 7UTC –8:45 UTC. But our retrievals suggest that plates occur more frequently than dendrite – 40% of our retrievals provide a better agreement between the measure and forward-simulated radar variables when using plates as the pristine ice habit, and 20% of retrievals have a better agreement using dendrite. That's why we conclude that our method does not distinguish plate and dendrites that well as we hoped.

We see the original text may be confusing, and have made some changes:

Lines 469–478:

... These expectations about prevalent ice habits are confirmed by visually checking the in-situ cloud particle images (see Fig. 1 for examples).

In our retrievals, 40% of the collocated radar observables are best fit with plate as the pristine ice habit, 20% with dendrite, and 40% with columns. In general, when the cloud particle images were dominated by columns, indeed, we have also found that retrievals with columns as the pristine ice habit provide the best agreement between the measured and forward-simulated radar observables. In the period between 7UTC –8:45 UTC when dendrites appeared much more frequently than plates in cloud particle images, our retrievals suggest the opposite, because 40% of best-fit retrievals are associated with plates and only 20% of best-fit retrievals are associated with dendrite. Therefore, we consider there remains a large uncertainty in distinguishing plate and dendrites using our retrievals. Note that even with this habit uncertainty, the choice of plate and dendrite does not lead to significantly different retrievals in $N_I$ and $q_I$.

30. Lines 509-510: How well collocated were the radar gates and the aircraft during these times? If the measurements are sampling regions of precipitation with different microphysical processes, of course the retrieval will not agree with the in situ measurements.

    Thanks for the question which leads to interesting findings. Responses are given in Comments #2.

31. Lines 518-519: Reflectivity also provides information about the pristine properties, especially in cases where vapor deposition dominates and aggregation is limited. Please rephrase.

    We have rephrased the sentence to:

    Lines 547–549:
    The first observable provides constraints on the combined aggregate and pristine ice population, while the last three observables provide constraints on the partitioning between aggregates and pristine ice, as well as on the ice number concentration and size of pristine ice.

32. Line 555: Add reference here for these equations.

    Fixed. We now have revised the text as:

    Line 595:
    Radar equations for a single sample volume containing multiple ice particle habits are given as (Jung et al., 2010):

**Response to Reviewer #2**

1. Microphysical retrievals of radar volumes containing a mixture of pristine ice particles and aggregates are challenging, since larger particles tend to dominate the signal. The manuscript presents a method for retrieving PSD parameters separately for crystals and aggregates from polarimetric radar observables based on an ensemble retrieval framework. The framework is constructed using a prior PSD parameter distribution and forward modeled radar observables from the assumed PSD based on scattering database results of a number of different kinds of pristine crystals and aggregates. The method is evaluated first with synthetic observations and then against in-situ aircraft measurements. The in-situ comparisons show an overall improvement over existing methods.

   The text is generally well written and structured. It involves adequate analysis and discussion of related uncertainties and the figures are clear and demonstrative. I expect the presented method to help advance the use of radar polarimetry in studying snow microphysics. I recommend the manuscript to be accepted for publication with minor revisions.

   We thank the reviewer for thoughtful comments.

*General comments*

2. My only general comment is that I would have liked to see some discussion related to the possibility of taking rimed particles in to account in similar retrievals. Riming may have great significance depending on climate and is expected to have a very similar polarimetric radar fingerprint as aggregation. Do you expect that riming could have affected your evaluation results?

   Some modifications need to be made for the retrieval method to work for rimed particles. We have provided our thoughts in the revised manuscript:

   > Line 575–582:
   >
   > Currently, our method is designed to work for conditions with a mixture of pristine ice and aggregates. In the presence of rimed particles, the state vector should be expanded to include additional variables that can accommodate and inform the degree of rimming, e.g., the riming factor described in Masson et al. (2018), or to include appropriate rimed species explicitly. When triple-frequency measurements are available and can be used to distinguish particle types effectively (e.g., Kneifel et al., 2015; Barrett et al., 2019), such information on particle types can also be incorporated into our method to provide retrievals for off-zenith radar scans that are more challenging for triple-frequency techniques. It is also possible to expand the observation vector with other radar observables at multiple wavelengths, providing further constraints on retrieval if added information exists.

*Specific comments*

3. 77: Instead of "spheroidal morphology" did you mean to make a statement on the aspect ratios of aggregates?

We meant to talk about their shape and orientation. For clarity, we have replaced "spheroidal morphology" with the following:

Lines 75–76:
Snow aggregates yield low $Z_{DR}$ (about 0–0.6 dB; see Hogan et al., 2012) as a result of their sparse and irregular morphology, with the component crystals oriented at a wide range of angles.

Lines 739–740:
Hogan, R. J., Tian, L., Brown, P. R. A., Westbrook, C., Heymsfield, A. J. and Eastment, J. D.: Radar scattering from ice aggregates using the horizontally aligned oblate spheroid approximation. J. Appl. Meteorol. Clim., 51, 655–671, doi: https://doi.org/10.1175/JAMC-D-11-074.1, 2012.

4. 86: Did you mean that these variables are simply less widely adopted or that there is more work to be done connecting characteristics in the retrievals of these variables to snow processes? Please rephrase

We apologize for the confusing wording. We meant that $\rho_{HV}$ and $K_{DP}$ observations have not been used extensively in quantitative retrievals. Since the other reviewer also has the same question, and the information provided here was not critical, we have removed this sentence.

5. 451-452: It is not evident to the reader what kind of temperature dip we are talking about since it seems to be excluded from the figure and not described here.

Apologies. We excluded the dip in the figure, so have reworded the text to explain the data gap:

Lines 466–468:
The flight height was maintained at ~2 km from 6:30 to 6:40 UTC, suggesting that the missing temperatures due to a data glitch at ~6:40 UTC are likely to be about –5°C.

6. 510: I'm not sure if I understood this sentence. Did you mean that these radar signatures might represent only a subset of the aircraft-collected sample? Or that there might be a spatial mismatch? Please rephrase and discuss the possible implications.

Thanks for pointing out the possibility of a spatial mismatch, which has helped us to look more carefully about the collocation.

To understand how well the collocation is, we have produced 2D histograms of occurrences of the vertical and horizontal distance in the collocated in-situ and radar dataset, as shown in Fig. R2. The distance was calculated with respect to radar gate, i.e., the positive vertical distance represents that the flight altitude is higher than the radar gate of interest.

Checking the collocation in Fig. R2, we found that in-situ samples were taken largely at radar scan heights or below in Clusters 1, 2 and 6. It is likely that both in-situ and radar have sampled the same regime with notable aggregations, and that's why the observed $D_{eff}$ is close to the retrieved $D_{eff}$ of aggregates. In contrast, in-situ samples were taken at higher altitudes over the radar scans in

Clusters 3–5. We believe that in Clusters 3–5, aircraft may have sampled a pristine ice growth zone aloft, but the radar gates below sampled the subsequent aggregations, which explains why the observed $D_{eff}$ is closer to the retrieved $D_{eff}$ of pristine ice, rather than aggregates.

In the revised manuscript, we now have included Fig. R2 as Fig. 11 and added the following text:

Lines 528–536:
The third scenario is that the discrepancy in $D_{eff}$ is due to a sampling issue. Figure 11 shows two-dimensional histograms of occurrences of the vertical and horizontal distance in the collocated in-situ and radar dataset. The distance was calculated with respect to radar gate, i.e., the positive vertical distance represents that the flight altitude is higher than the radar gate of interest. Interestingly, for Clusters 1, 2 and 6, in-situ samples were taken largely at radar scan heights or below. It is likely that both in-situ and radar have sampled the same regime with notable aggregations, which explains why the observed $D_{eff}$ is close to the retrieved $D_{eff}$ of aggregates. In contrast, in-situ samples were taken at higher altitudes over the radar scans for Clusters 3–5. In these cases, aircraft may have sampled a pristine ice growth zone aloft, but the radar gates below sampled the subsequent aggregations, which explains why the observed $D_{eff}$ is closer to the retrieved $D_{eff}$ of pristine ice, rather than aggregates. Further studies using more datasets and retrievals would be needed to assess the third scenario.

Lines 569–574 in the Summary Section:
… In other clusters, the observed effective mean diameters agree better with the retrieved size of pristine ice, likely because the aircraft sampled pristine ice growth zones aloft instead of aggregation zones that radar sampled. Since planar crystal growth and subsequent aggregation can lead to zones with distinct ice bulk properties, taking frequent aircraft measurements at multiple vertical layers around the radar location would be particularly helpful to improve collocations and allow us to analyse individual rays in more detail.

[Figure]

**Figure R2:** *2D histograms of occurrences of distances in the vertical and horizontal between in-situ measurements and radar gates for Cluster 1–6 in (a)–(f), respectively. Note that occurrences are counted for all pairs of in-situ data point and radar gate. In calculations of retrieval errors, selected in-situ data points and radar gates are only used once with equal weights.*

*Technical comments*

7. 424-426: This could be rephrased to avoid repetition.

   Thank you.  We have rephrased it to the following:
* * *
   Lines 437–438:
   (k) and (l) represent the individual and combined effective mean diameters using the maximum particle dimension and the equivalent melted particle as the size descriptor, respectively.
* * *
8. 483: (f) should be (d).

   Done – corrected (f) to (d) on Line 501.

9. 512: particles

   The sentence (on Line 536) has been rewritten, and thus this error is removed.